# Mesolimbic dopamine projections mediate cue-motivated reward seeking but not reward retrieval in rats

**Briac Halbout[1,2]\*, Andrew T Marshall[1,2], Ali Azimi[1,2], Mimi Liljeholm[3], Stephen V Mahler[2,4], Kate M Wassum[5,6], Sean B Ostlund[1,2]\***

[1]Department of Anesthesiology and Perioperative Care, University of California, Irvine, Irvine, United States; [2]Irvine Center for Addiction Neuroscience, University of California, Irvine, Irvine, United States; [3]Department of Cognitive Sciences, University of California, Irvine, Irvine, United States; [4]Department of Neurobiology and Behavior, University of California, Irvine, Irvine, United States; [5]Department of Psychology, University of California, Los Angeles, Los Angeles, United States; [6]Brain Research Institute, University of California, Los Angeles, Los Angeles, United States

**Abstract** Efficient foraging requires an ability to coordinate discrete reward-seeking and reward-retrieval behaviors. We used pathway-specific chemogenetic inhibition to investigate how rats' mesolimbic and mesocortical dopamine circuits contribute to the expression and modulation of reward seeking and retrieval. Inhibiting ventral tegmental area dopamine neurons disrupted the tendency for reward-paired cues to motivate reward seeking, but spared their ability to increase attempts to retrieve reward. Similar effects were produced by inhibiting dopamine inputs to nucleus accumbens, but not medial prefrontal cortex. Inhibiting dopamine neurons spared the suppressive effect of reward devaluation on reward seeking, an assay of goal-directed behavior. Attempts to retrieve reward persisted after devaluation, indicating they were habitually performed as part of a fixed action sequence. Our findings show that complete bouts of reward seeking and retrieval are behaviorally and neurally dissociable from bouts of reward seeking without retrieval. This dichotomy may prove useful for uncovering mechanisms of maladaptive behavior.
DOI: https://doi.org/10.7554/eLife.43551.001

\*For correspondence:
halboutb@uci.edu (BH);
sostlund@uci.edu (SBO)

## Introduction

Foraging and other reward-motivated behaviors tend to unfold as a sequence of actions, beginning with a *reward-seeking* phase and ending with an attempt to retrieve and consume any rewards produced by this activity. Coordinating the discrete reward-seeking and reward-retrieval behaviors that make up these action sequences is important for efficient foraging. When rewards are sparse or otherwise difficult to obtain, attempts to retrieve them are often unnecessary and should therefore be withheld to conserve energy and minimize opportunity costs (*Stephens and Krebs, 1986*; *Niv et al., 2007*). Consistent with this, studies on self-paced instrumental behavior show that the ability to efficiently pattern reward-seeking and -retrieval responses based on task demands (e.g., reinforcement schedule) can strongly impact the rate at which rewards are obtained (*Ostlund et al., 2012*; *Wassum et al., 2012*; *Matamales et al., 2017*). However, such behaviors must remain sensitive to changes in internal and external states. For instance, environmental cues that signal reward availability increase attempts to seek out (*Estes, 1948*; *Corbit and Balleine, 2016*) and retrieve reward (*Marshall and Ostlund, 2018*). While the ability to develop and modify action sequences is normally adaptive, this process may become dysregulated in certain conditions, such as obsessive-compulsive disorder (*Joel and Avisar, 2001*; *Korff and Harvey, 2006*; *Frederick and Cocuzzo, 2017*) and drug

addiction (*Tiffany, 1990*; *Graybiel, 2008*; *Volkow et al., 2013*), leading to maladaptive behaviors. Despite this, the behavioral and neural mechanisms responsible for regulating reward seeking and retrieval are not well understood.

Previous studies strongly implicate dopamine in learning new action sequences (*Graybiel, 1998*; *Jin and Costa, 2015*). While other findings suggest that dopamine is not as important for the expression of well-established action sequences (*Levesque et al., 2007*; *Wassum et al., 2012*), it remains possible that dopamine contributes to action sequence performance when changes in task conditions prompt a reorganization of reward seeking and retrieval. For instance, previous studies indicate that the tendency for reward-paired cues to motivate reward-*seeking* behavior critically depends on dopamine signaling (*Dickinson et al., 2000*; *Ostlund and Maidment, 2012*; *Wassum et al., 2011*), particularly in the nucleus accumbens (NAc) (*Wyvell and Berridge, 2000*; *Lex and Hauber, 2008*; *Wassum et al., 2013*; *Ostlund et al., 2014*; *Aitken et al., 2016*). Interestingly, we recently found that such cues do not simply provoke reward-seeking behavior (e.g., lever pressing), they also increase the likelihood that such behavior will be followed by an attempt to retrieve reward (e.g., food-cup approach)(*Marshall and Ostlund, 2018*). Although this finding suggests that reward-paired cues preferentially motivate complete bouts of reward seeking and retrieval, it has yet to be established if this modulation of action sequence performance depends on dopamine.

Dopamine may also contribute to regulating attempts to seek out and retrieve a reward when the value of that reward changes. Self-paced, instrumental reward-seeking actions are normally performed in a goal-directed manner, such that they are sensitive to changes in reward value (*Balleine and Dickinson, 1998*). However, they can develop into inflexible stimulus-response habits with extended training (*Dickinson, 1985*; *Dickinson et al., 1995*). In contrast, it is not well understood how changes in reward value modulate attempts to retrieve rewards produced through instrumental reward-seeking behavior. For example, it has been suggested that rats' tendency to approach the food cup after lever pressing may represent a discrete goal-directed action – one that is selected independently of the initial decision to press the lever (*Rescorla, 1964*). Alternatively, rats may concatenate the press-approach sequence to form an *action chunk*, which can then be selected and deployed as a single unit of behavior (*Lashley, 1951*; *Graybiel, 1998*; *Jin and Costa, 2015*). Action chunks are thought to represent a special form of habit, or behavioral chain, in which each element of the chain automatically elicits the next response. This allows for efficient action sequencing but comes with a decrease in behavioral flexibility. Once an action chunk has been initiated, it should be automatically completed without further consideration of reward value (*Dezfouli et al., 2014*; *Smith and Graybiel, 2016*).

In the current study, we applied a chemogenetic approach to investigate the role of the mesocorticolimbic dopamine system in action sequence performance in rats. We used a combination of well-established behavioral assays and novel microstructural analyses to selectively probe the influence of reward-paired cues and expected reward value on the regulation of reward-seeking and -retrieval responses. We found that inhibiting dopamine neurons in the ventral tegmental area (VTA) or their inputs to the NAc, but not the medial prefrontal cortex (mPFC), reversibly disrupted cue-motivated reward seeking, but spared the tendency for reward-paired cues to trigger complete bouts of seeking and retrieval. These dopamine manipulations had no impact on rats' tendency to adjust their reward-seeking behavior in response to reward devaluation. Importantly, attempts to retrieve reward were not suppressed by reward devaluation, suggesting that this behavior was the product of action chunking.

## Results

### Effects of response-contingent feedback about reward delivery on reward retrieval

We first characterized the relationship between reward-seeking and -retrieval responses when rewards are sparse (*Figure 1A*). Rats were trained to lever press on a RI-60s schedule, such that this action was often nonreinforced and only occasionally earned food pellet delivery into a recessed food cup. Not surprisingly, we found that the probability of food-cup approach was elevated for several seconds after performance of the lever-press action (*Figure 1B and C*). This timeframe for

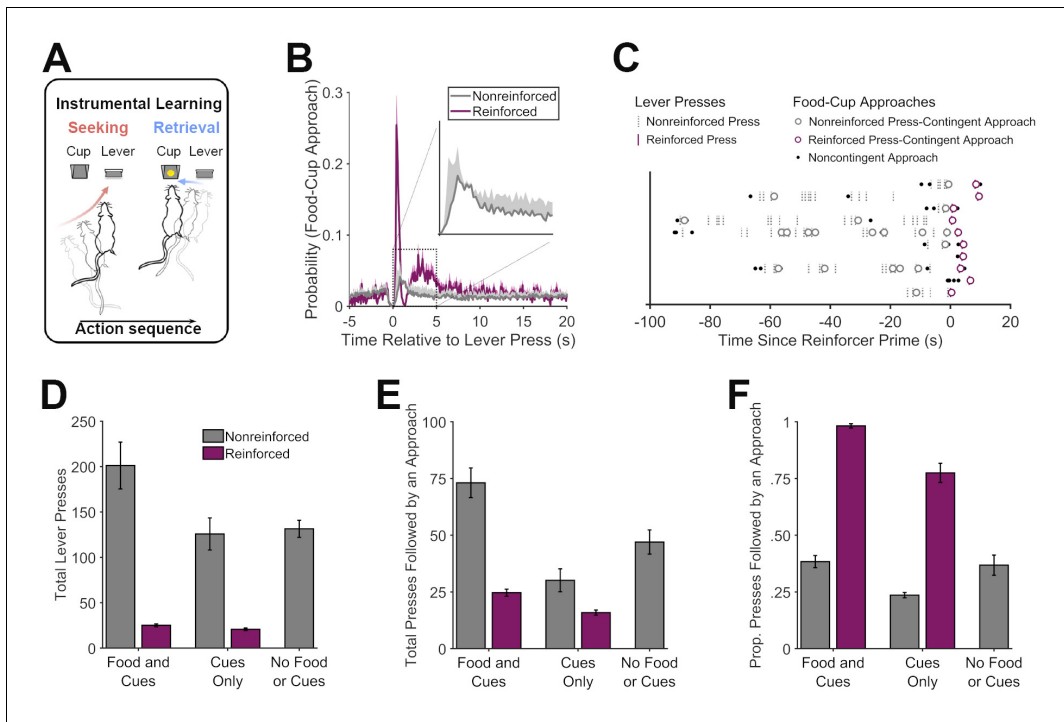

**Figure 1.** Microstructural organization of instrumental behavior. (**A**) Hungry rats were trained to perform a self-paced 'reward seeking' task, in which pressing a lever was intermittently reinforced with food pellets (RI-60s schedule). Press-contingent food-cup approaches were taken as a measure of attempted 'reward retrieval'. (**B**) Probability of food-cup approaches as a function of time surrounding reinforced (purple) and nonreinforced (gray) lever presses. (**C**) Representative pattern of food-cup approach behavior for an individual rat surrounding reinforced and nonreinforced lever presses. Individual reinforced trials are separately presented across the y-axis aligned at the point at which the lever became activated (i.e., primed for reinforcement). (**D, E**) Effects of manipulating instrumental reinforcement contingency on the organization of reward-seeking and -retrieval responses. Total lever presses (**D**) or presses followed by an approach (**E**) during tests in which lever pressing was intermittently reinforced (RI-60s) either with food pellets and associated cues (Food and Cues) or with pellet dispenser cues but no actual food delivery (Cues Only). Rats were also tested without any reinforcement (No Food or Cues). (**F**) The proportion of lever presses that were followed by food-cup approach was higher for reinforced presses than for nonreinforced presses, regardless of whether pressing was reinforced with Food and Cues, or Cues Only. Rats also continued to sporadically check the food cup after nonreinforced lever presses, albeit at a much lower level than after reinforced presses.

DOI: https://doi.org/10.7554/eLife.43551.002

The following source data is available for figure 1:

**Source data 1.** This spreadsheet contains the behavioral responses for individual rats in *Figure 1*.
DOI: https://doi.org/10.7554/eLife.43551.003

press-contingent food-cup approach behavior is consistent with previous reports (*Nicola, 2010*; *Marshall and Ostlund, 2018*), and was relatively consistent across the current experiments (see *Figure 3—figure supplement 1*). We therefore used a cutoff value of 2.5 s to identify reward-retrieval attempts. To control for reward-retrieval *opportunities*, which were contingent on lever pressing, our analysis focuses on a normalized measure – the proportion of lever presses that were followed by food-cup approach.

We found that rats were much more likely to approach the food cup after reinforced presses than after nonreinforced presses ($t(8) = 19.33$, $p<0.001$), suggesting they could detect when pellets were delivered based on sound and tactile cues produced by the dispenser. This was confirmed in subsequent tests, during which lever pressing produced either 1) pellet dispenser cues and actual pellet delivery (Food and Cues), 2) pellet dispenser cues only (Cues Only), or 3) no pellet dispenser cues or pellet delivery (No Food or Cues). Here too, we found that food-cup approaches were more likely

after reinforced than nonreinforced lever presses, regardless of whether pellet dispenser cues were presented alone or together with actual food delivery (*Figure 1F*; $ts(8) \geq 13.74$, $ps <0.001$; the overall frequency of lever pressing (*Figure 1D*) and the frequency of complete bouts of presses that were followed by an approach (*Figure 1E*) are presented for comparison). Although pellet dispenser cues were clearly an effective trigger for rats to shift from the lever to the food cup, they also made these shifts spontaneously, indicating that they had developed the tendency to perform the complete press-approach action sequence. These unprompted approaches occurred after a relatively small subpopulation of nonreinforced lever presses, which is consistent with our previous data (*Marshall and Ostlund, 2018*).

### Inhibiting dopamine neurons during Pavlovian-to-instrumental transfer preferentially disrupts cue-motivated reward seeking, but not reward retrieval

Our previous findings suggest that reward-predictive cues *both* invigorate reward-seeking behavior (i.e., the PIT effect) and increase the likelihood that such actions will be followed by an attempt to retrieve reward from the food cup (*Marshall and Ostlund, 2018*). Experiment 2 investigated the contributions of the mesocorticolimbic dopamine system to these distinct behavioral effects of reward-paired cues.

Rats with dopamine neuron-specific expression of the inhibitory DREADD hM4Di or mCherry in the VTA (*Figure 2*) were trained on a PIT task (*Figure 3A*) consisting of a Pavlovian conditioning phase, in which two different auditory cues were paired (CS+) or unpaired (CS-) with food pellets, and a separate instrumental training phase, in which rats were trained to lever press for pellets. During PIT testing, we noncontingently presented the CS+ and CS- while rats were free to lever press and check the food cup without response-contingent food or cue delivery.

We found that rats selectively increased their lever press performance during CS+ presentations, relative to the CS- and pre-CS response rates (*Figure 3B*; CS Period * CS Type interaction, p<0.001; see *Supplementary file 1A* for full generalized linear mixed-effects model output). This effect was significantly attenuated by CNO in a group-specific manner (Group * Drug * CS Period * CS Type interaction, p=0.002). Analysis of data from CS+ trials (only) found that CNO selectively suppressed cue-induced lever pressing in hM4Di relative to mCherry rats (Drug * Group * CS Period interaction, p=0.013). Further analysis found that the mCherry group displayed a pronounced increase in lever pressing during CS+ trials (CS Period * CS Type interaction, p<0.001), and this effect was not altered

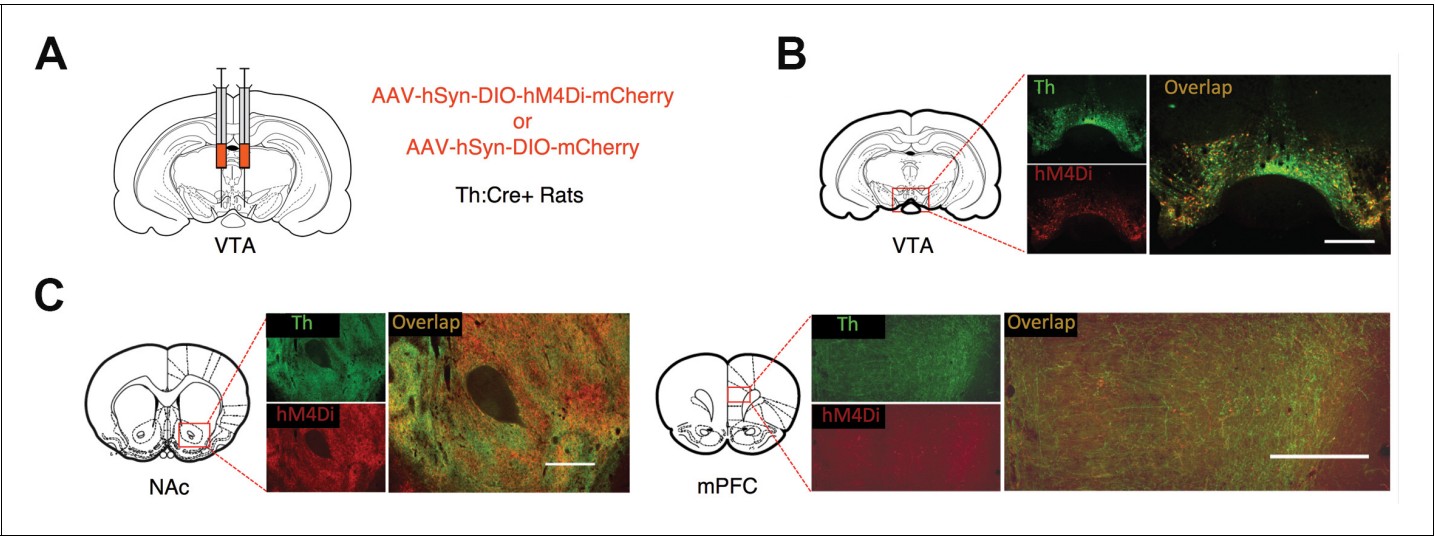

**Figure 2.** DREADD expression in Th:Cre +rats. (**A**) Th:Cre+ rats received bilateral injections of AAV-hSyn-DIO-hM4Di-mCherry or AAV-hSyn-DIO-mCherry in the VTA. (**B**) Representative expression of the mCherry-tagged inhibitory DREADD hM4Di (red) in VTA Th positive neurons (green) of Th:Cre + rats, as well as in neuronal terminals (**C**) projecting to the nucleus accumbens (NAc) and medial prefrontal cortex (mPFC). Scale bar is 500 μm.
DOI: https://doi.org/10.7554/eLife.43551.004

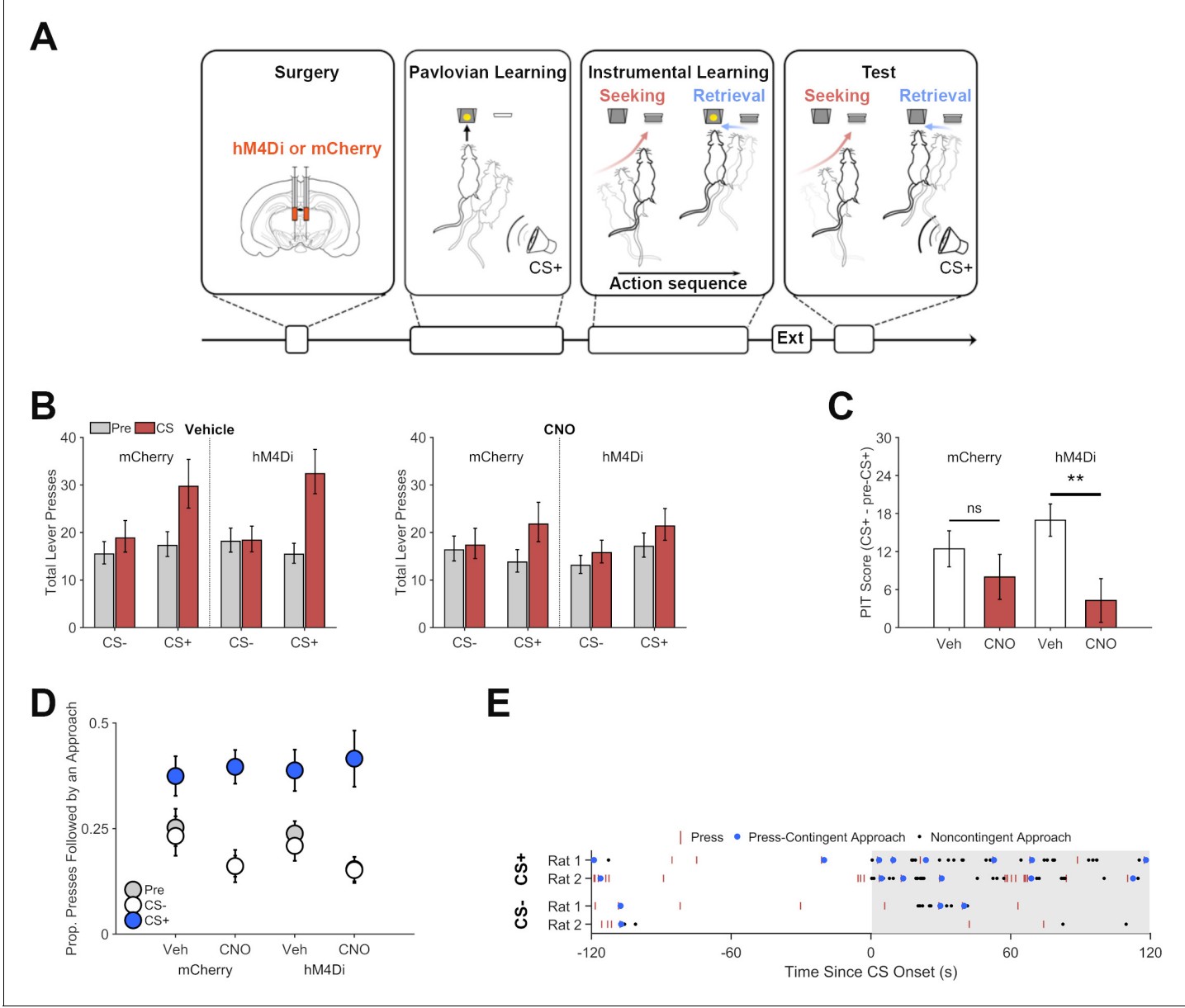

**Figure 3.** Chemogenetic inhibition of dopamine neurons on Pavlovian to instrumental transfer (PIT) performance. (**A**) Experimental design: Following viral vectors injections and recovery, rats received Pavlovian training, during which they learned to associate an auditory cue (CS+) with food pellet delivery. During instrumental conditioning, rats performed the same lever-press task used in Experiment 1. Lever pressing was extinguished (Ext) before rats were submitted to a PIT test, which included separate noncontingent presentations of the CS+ and an unpaired control cue (CS-). (**B**) Chemogenetic inhibition of VTA dopamine neurons disrupted cue-motivated reward seeking. Total lever presses during PIT trials for rats expressing the inhibitory DREADD hM4Di or mCherry following vehicle (left) or CNO (5 mg/kg, right) treatment prior to test. Presses during pre-CS (gray) and CS periods (red) are plotted separately. (**C**) PIT expression is specifically impaired in hM4Di expressing Th:Cre+ rats. PIT scores (total presses: CS+ - pre-CS+) show that the CS+ increased lever pressing after vehicle treatment for both groups, but that CNO suppressed this effect in the hM4Di group but not the mCherry group. **p<0.01. (**D**) The CS+ increased the proportion of lever presses that were followed by a food-cup approach during PIT testing. Inhibiting VTA dopamine neurons did not disrupt expression of this effect. Instead, rats in both groups showed a modest increase in their likelihood of checking the food cup after lever pressing when treated with CNO. (**E**) Representative organization of the effects of the CS+ and CS- on attempts to seek out and retrieve reward during PIT. Data show lever presses and food-cup approaches (press-contingent or noncontingent) for two control rats (Th:Cre+ rats expressing mCherry and receiving vehicle).
DOI: https://doi.org/10.7554/eLife.43551.005

The following source data and figure supplements are available for figure 3:

**Source data 1.** This spreadsheet contains the behavioral responses for individual rats in *Figure 3*.

*Figure 3 continued*

DOI: https://doi.org/10.7554/eLife.43551.009

**Figure supplement 1.** Probability of food-cup approaches as a function of time surrounding individual lever-press responses during PIT testing, plotted separately for CS+ (blue), CS- (red), and pre-CS (yellow) periods.

DOI: https://doi.org/10.7554/eLife.43551.006

**Figure supplement 2.** Frequency of lever presses that were followed by a food-cup approach during PIT testing by rats expressing the inhibitory DREADD hM4Di or mCherry following vehicle or CNO (5 mg/kg) treatment in Experiment 2.

DOI: https://doi.org/10.7554/eLife.43551.007

**Figure supplement 3.** Noncontingent (press-independent) food-cup approaches during PIT testing by rats expressing the inhibitory DREADD hM4Di or mCherry following vehicle or CNO (5 mg/kg) treatment in Experiment 2.

DOI: https://doi.org/10.7554/eLife.43551.008

by CNO (Drug * CS Period * CS Type interaction, p=0.780). In contrast, CNO pretreatment significantly disrupted expression of CS+ induced lever pressing in the hM4Di group (Drug * CS Period * CS Type interaction, p<0.001). hM4Di rats showed a CS+ specific elevation in lever pressing when pretreated with vehicle (CS Period * CS Type interaction, p<0.001) but not CNO (CS Period * CS Type interaction, p=0.684). While these findings indicate that CNO selectively disrupted the response-invigorating influence of the CS+ by inhibiting VTA dopamine neurons in hM4Di rats, there was also some indication that CNO may have produced a nonspecific, group-independent, suppression of PIT performance (Drug x CS Period x CS Type, p=0.007). We therefore conducted a more focused analysis of CS+ induced changes in lever-press performance (PIT score: CS+ - pre-CS+; *Figure 3C*), which confirmed that CNO significantly suppressed this behavioral effect in the hM4Di group (t(17) = −3.83, p<0.001), but not in the mCherry group (t(13) = −1.21, p=0.249). This is in line with recent findings that similar CNO treatment does not significantly alter PIT performance in DREADD-free rats (*Collins et al., 2019*).

We also investigated if VTA dopamine neuron inhibition impacts the tendency for the CS+ to increase attempts to retrieve reward after performing the reward-seeking response (*Figure 3D and E*; see *Figure 3—figure supplement 1* for illustration of the probability of food-cup approach surrounding lever presses during nonreinforced PIT trials). We found that the CS+ (p<0.001) but not the CS- (p=0.501) increased the proportion of lever presses that were followed by a food-cup approach, even though no rewards were actually delivered at test (see *Supplementary file 1B* for full generalized linear mixed-effects model output; see *Figure 3—figure supplements 2* and *3* for analysis of total press-contingent and noncontingent approaches, respectively). Importantly, CNO did not alter this response to the CS+ in a group-specific manner (Group * Drug * CS+ Period, p=0.835), indicating that VTA dopamine neuron function is not required for this behavior. However, CNO did induce some nonspecific, group-independent alterations in the proportion of presses that were followed by a food-cup approach, lowering the overall likelihood of this behavior (Drug effect, p=0.019), but *enhancing* the effect of the CS+ (Drug * CS+ Period, p<0.037).

## Pathway-specific inhibition of dopamine projections to NAc, but not mPFC, disrupts cue-motivated reward seeking but not retrieval

As previously reported (*Mahler et al., 2019*), hM4Di expression in VTA dopamine neurons resulted in transport of DREADDs to axonal terminals in the NAc and mPFC (*Figure 2*). We took advantage of this to investigate the roles of these two pathways in PIT performance, again distinguishing between the influence of reward-paired cues on reward seeking and reward retrieval. Guide cannulae were aimed at the NAc or mPFC in rats expressing hM4Di in VTA dopamine neurons (Experiment 3A; *Figure 4A* and *Figure 4—figure supplement 1*). These rats underwent training and testing for PIT (*Figure 4B*), as described above, but were pretreated with intra-NAc or mPFC injections of CNO (1 mM) or vehicle to achieve local inhibition of neurotransmitter release (*Mahler et al., 2014*; *Stachniak et al., 2014*; *Lichtenberg et al., 2017*), an approach previously shown to be effective in inhibiting dopamine release (*Mahler et al., 2019*). *Figure 4C* shows that, in hM4Di-expressing rats, the CS+ specific increase in lever pressing (CS Period * CS Type interaction, p<0.001) was disrupted by CNO in a manner that depended on microinjection site (Drug * CS Period * CS Type * Site interaction, p=0.003; *Supplementary file 1C* for full generalized linear mixed-effects model output). After intracranial vehicle injections, rats showed a CS+ specific elevation in pressing (CS Period * CS

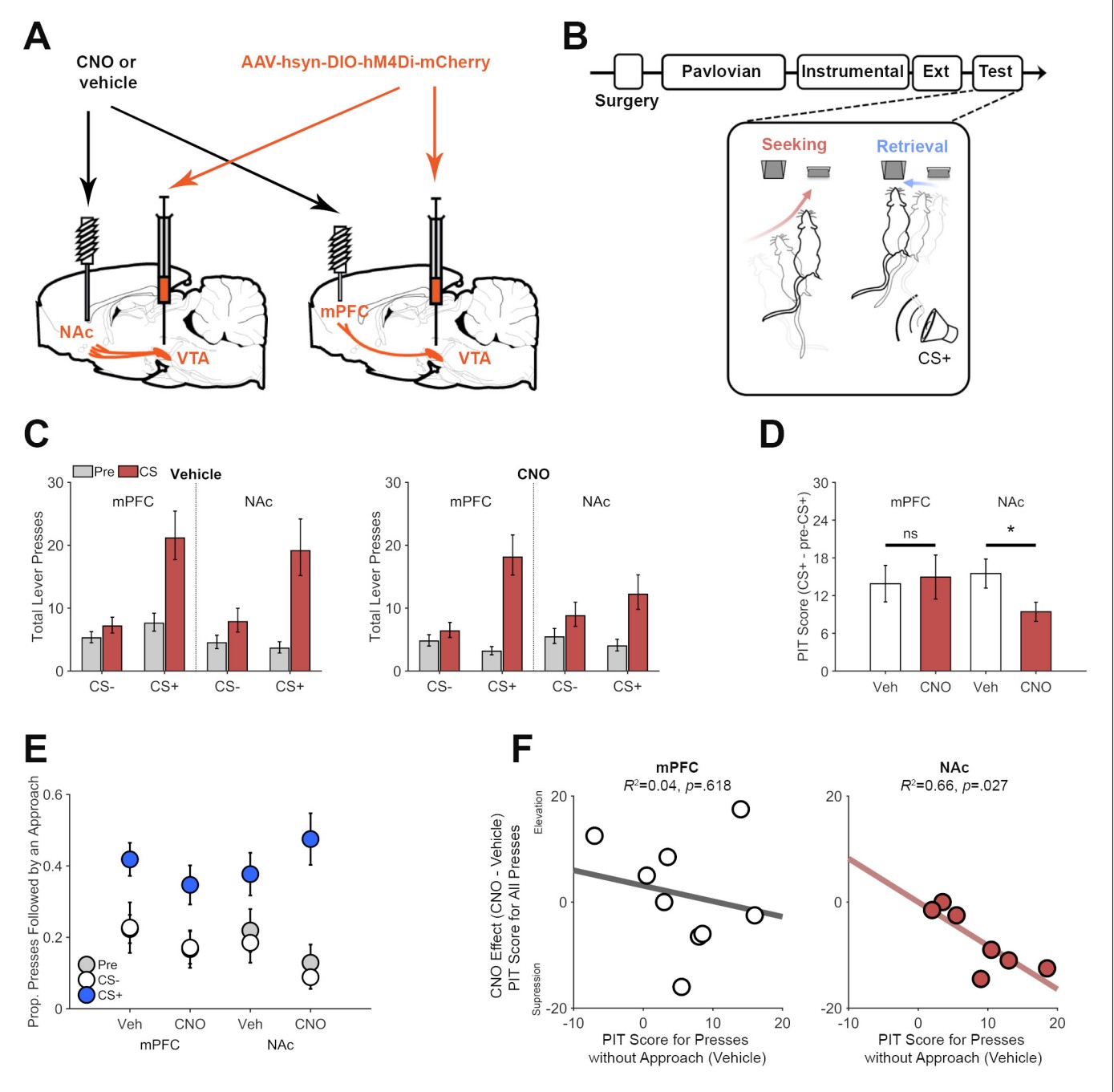

**Figure 4.** Pathway specific chemogenetic inhibition of dopamine on PIT performance.  (A) Th:Cre+ rats initially received VTA AAV-hSyn-DIO-hM4Di-mCherry injections and were implanted with guide cannulas aimed at the medial prefrontal cortex (mPFC) or nucleus accumbens (NAc) for microinjection of CNO (1 mM) or vehicle to inhibit dopamine terminals at test. (B) Following surgery, rats underwent training and testing for PIT, as described above. We analyzed the microstructural organization of behavior (Lever presses: seeking, and presses followed by a food-cup approach: retrieval) at test. (C) Pathway specific inhibition of dopamine terminals in the NAc but not the mPFC disrupted cue-motivated reward seeking. Total lever presses during PIT trials for rats expressing the inhibitory DREADD hM4Di and receiving CNO or vehicle microinfusions in either the mPFC or NAc prior to test. Presses during pre-CS (gray) and CS periods (red) are plotted separately. (D) PIT expression was specifically impaired following NAc CNO treatment. PIT scores (total presses: CS+ - pre-CS+) show that the CS+ increased lever pressing following vehicle treatment in both groups, but that CNO suppressed this effect when injected into the NAc but not the mPFC. *p<0.05. (E) The CS+ increased the proportion of lever presses that were followed by a food-cup approach during PIT testing. This effect did not significantly vary as a function of drug treatment or group. (F) Scatter plots show the relationship between individual differences in the effect of the CS+ on lever presses that were not followed by food-cup approach in the

*Figure 4 continued on next page*

*Figure 4 continued*

vehicle condition (PIT Score for presses without approach) and the suppressive effect of CNO on CS+ evoked lever pressing (PIT Score for CNO test - PIT Score for vehicle test). Data points are from individual rats receiving intra-mPFC (left panel) or intra-NAc (right panel) microinjections.

DOI: https://doi.org/10.7554/eLife.43551.010
The following source data and figure supplements are available for figure 4:

**Source data 1.** This spreadsheet contains the behavioral responses for individual rats in *Figure 4*.
DOI: https://doi.org/10.7554/eLife.43551.016
**Figure supplement 1.** Cannulae placements for Experiment 3A hM4Di expressing rats.
DOI: https://doi.org/10.7554/eLife.43551.011
**Figure supplement 2.** Frequency of lever presses during PIT testing by rats expressing mCherry following microinjection of vehicle (A) or CNO (B) into the mPFC or NAc in Experiment 3B.
DOI: https://doi.org/10.7554/eLife.43551.012
**Figure supplement 3.** Frequency of presses that were followed by a food-cup approach during PIT testing by rats expressing the inhibitory DREADD hM4Di following microinjection of CNO or vehicle into the mPFC or NAc in Experiment 3A.
DOI: https://doi.org/10.7554/eLife.43551.013
**Figure supplement 4.** Noncontingent (press-independent) food-cup approaches during PIT testing in rats expressing the inhibitory DREADD hM4Di following microinjection of CNO or vehicle into the mPFC or NAc in Experiment 3A.
DOI: https://doi.org/10.7554/eLife.43551.014
**Figure supplement 5.** Scatter plots show the relationship between individual differences in the effect of the CS+ on lever presses that were not followed by food-cup approach in the vehicle condition (PIT score for presses without approach) and the suppressive effect of CNO on CS+ evoked lever pressing (PIT Score for CNO test - PIT Score for vehicle test).
DOI: https://doi.org/10.7554/eLife.43551.015

Type interaction, p<0.001), which did not differ significantly across vehicle injection sites (CS Period * CS Type * Site interaction, p=0.151). Unlike with systemic CNO, the CS+ remained effective in increasing lever pressing after CNO microinjection into the mPFC (CS Type * CS Period interaction, p<0.001) and NAc (CS Type * CS Period interaction, p<0.001). However, this effect was significantly attenuated when CNO was injected into the NAc versus the mPFC (CS Period * CS Type * Site interaction, p=0.012; analysis of CNO data only). A more focused analysis of CS+ elicited lever pressing (*Figure 4D*; PIT score) confirmed that CNO disrupted this effect in the NAc group ($t(6) = -2.49$, p=0.047), but not in the mPFC group ($t(8) = 0.34$, p=0.746).

The disruptive effect of intra-NAc CNO administration on PIT performance did not systematically vary as a function of injection site (data not presented), which is not surprising given previous findings that this effect is modulated by dopamine signaling in both the core and shell of the NAc (*Lex and Hauber, 2008*; *Peciña and Berridge, 2013*). Given such findings, it is possible that complete inhibition of ventral striatal dopamine transmission would abolish expression of the PIT effect, as it was found with systemic CNO treatment in Experiment 2. It is also possible that VTA dopamine projections to areas not targeted in the current study (e.g., amygdala) make an important, parallel contribution to this behavior.

We also conducted a separate experiment (Experiment 3B) with rats expressing the mCherry reporter (only) in VTA dopamine neurons to determine if this behavioral effects of CNO microinfusion was hM4Di-dependent. While there was evidence that CNO may have produced some nonspecific response suppression when injected into the mPFC but not the NAc (Drug * Site * CS Period * CS Type, p=0.068), this drug treatment did not significantly disrupt expression of CS+ elicited lever pressing for either injection site (*p*'s > 0.165; *Figure 4—figure supplement 2*).

As in the previous experiment, we found that the CS+ (p<0.001) increased the proportion of lever presses that were followed by an attempt to retrieve reward from the food cup (*Figure 4E*; *Supplementary file 1D* for full generalized linear mixed-effects model output; see *Figure 4—figure supplements 3* and *4* for analysis of total press-contingent and noncontingent approaches, respectively). CNO seemed to generally reduce the likelihood that lever pressing would be followed by food-cup approach, though this effect did not reach statistical significance (Drug effect, p=0.057). If anything, intra-NAc injections of CNO tended to *enhance* the effect of the CS+ on this approach response, though this effect also failed to reach significance (Drug * Site * CS+ Period, p=0.093).

The above findings indicate that VTA dopamine circuitry supports the motivational influence of the CS+ on reward seeking but does not mediate that cue's ability to promote reward retrieval. We

wondered if this might account for variability in the partial, response-suppressive effect of NAc dopamine terminal inhibition. Specifically, we hypothesized that rats inclined to respond to the CS + by engaging in discrete bouts of lever pressing, without attempting to retrieve reward, would be particularly sensitive to inhibition of NAc dopamine inputs. Consistent with this, we found that for the NAc group, individual differences in the effect of the CS+ on lever presses without subsequent food cup approach (during the vehicle test) were correlated with the degree to which CNO suppressed CS+ evoked lever pressing (PIT Score for all presses), relative to vehicle (CNO – Vehicle; $r = -0.81$, p=0.027; *Figure 4F*). No such relationship was found for the mPFC group ($r = -0.19$, p=0.618), which did not show sensitivity to dopamine terminal inhibition. Similar analysis of data from Experiment 2 also found no correlation between these measures (*Figure 4—figure supplement 5*), which may not be surprising given that systemic inhibition of VTA dopamine neurons led to a more robust and consistent suppression of CS+ evoked lever pressing (*Figure 3B*).

Altogether, these findings demonstrate that the mesolimbic dopamine system selectively mediates cue-motivated reward seeking, and suggest that dopamine inputs to the NAc are particularly important for individuals that tend to respond to such cues with discrete bouts of reward seeking without subsequent reward retrieval.

## Inhibiting dopamine neurons spares the sensitivity of reward-seeking actions to reward devaluation

It is unclear from the above findings if rats' tendency to approach the food cup after lever pressing reflects a discrete goal-directed action or if this response tends to be performed habitually, as part of a fixed press-approach action chunk. We conducted a reward devaluation experiment to probe this issue and investigate the role of VTA dopamine neurons in goal-directed action selection. Rats expressing mCherry or hM4Di in VTA dopamine neurons were trained on two distinct instrumental action-outcome contingencies, after which they underwent reward devaluation testing after pretreatment with CNO (5 mg/kg) or vehicle (*Figure 5A*). Rats performed significantly fewer presses on the devalued lever than on the valued lever (*Figure 5B*; Lever effect, p<0.001; *Supplementary file 1E* for full generalized linear mixed-effects model output). CNO treatment did not significantly alter the effect of reward devaluation on lever pressing in either hM4Di or mCherry rats (Drug * Lever, p=0.146; Group * Drug * Lever interaction, p=0.591), indicating that VTA dopamine neuron function is not required for this aspect of goal-directed action selection. Inhibiting VTA dopamine neurons also failed to disrupt sensitivity to devaluation during reinforced testing (see *Figure 5—figure supplement 1*).

VTA dopamine neuron inhibition did not significantly alter the overall likelihood of press-contingent approach behavior or its sensitivity to reward devaluation (*Figure 5C*; $ps \geq . 109$; see *Supplementary file 1F* for full generalized linear mixed-effects model output). Interestingly, we found that the proportion of presses that were followed by a food-cup approach was actually *greater* for the devalued lever than for the valued lever (Lever effect, p=0.040). This effect was driven by the fact that lever presses that were not followed by approach were more strongly suppressed by reward devaluation than presses that were directly followed by an approach (Press Type * Lever interaction, p<0.001; see *Figure 5D* and *Supplementary file 1G* for full generalized linear mixed-effects model output).

It was not possible to analyze the impact of reward devaluation on noncontingent approach responses performed at test because this behavior was associated with both the valued and devalued reward. However, other findings from our lab (data not shown) from studies involving a single reward-type indicate that noncontingent approaches are readily suppressed by reward devaluation, in contrast to response-contingent approaches. This is in line with previous reports that food-cup approach behavior is generally sensitive to reward devaluation (*Balleine, 1992*; *Thrailkill and Bouton, 2017*), particularly if it is elicited by Pavlovian reward-predicted cues (*Holland and Straub, 1979*; *Lichtenberg et al., 2017*).

## Discussion

We investigated the role of mesocorticolimbic dopamine circuitry in regulating reward-seeking (lever pressing) and reward-retrieval responses (press-contingent food-cup approach). Consistent with a recent study (*Marshall and Ostlund, 2018*), we found that noncontingent CS+ presentations

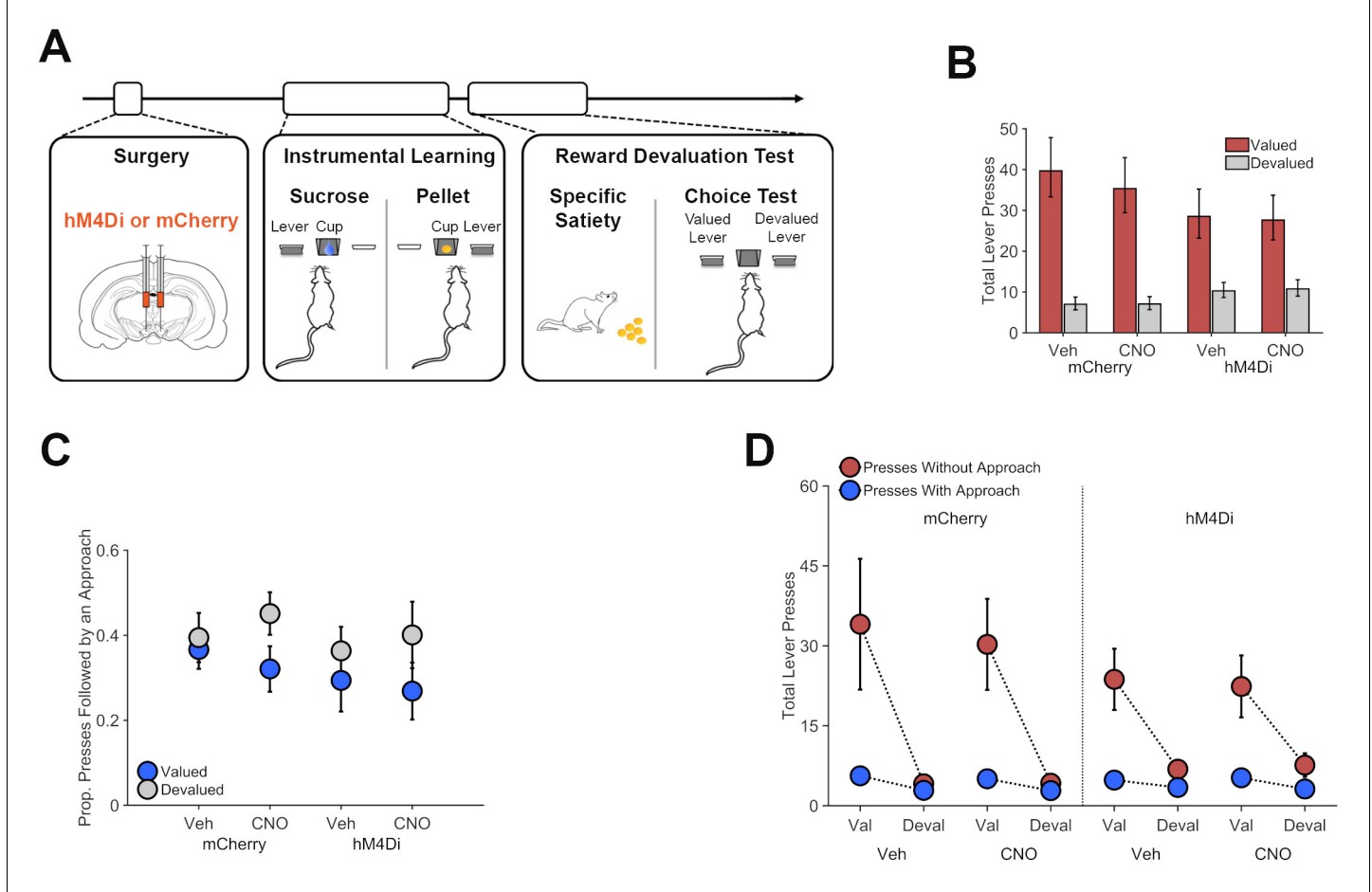

**Figure 5.** Chemogenetic inhibition of dopamine neurons on reward devaluation performance. (**A**) Th:Cre+ rats received VTA injections of AAV-hSyn-DIO-hM4Di-mCherry or AAV-hSyn-DIO-mCherry. Following recovery, rats were trained on two distinct lever-press actions for two different rewards (Instrumental Learning). Rats then underwent reward-specific devaluation testing following treatment with CNO (5 mg/kg) or vehicle. (**B**) Chemogenetic VTA dopamine inhibition did not alter the impact of reward devaluation on reward seeking. Total lever presses on the valued (red bars) and devalued (gray) levers in hM4Di or mCherry expressing Th:Cre+ rats, following CNO (5 mg/kg) or vehicle treatments. (**C**) Proportion of valued (blue) and devalued (gray) lever-press actions that were followed by a food-cup approach. Rats were more likely to attempt to retrieve reward after performing the devalued lever-press action. This effect was not altered by VTA dopamine neuron inhibition. (**D**). Lever presses performed without a subsequent food-cup approach response (red) were more sensitive to reward devaluation than presses that were followed by an approach (blue).

DOI: https://doi.org/10.7554/eLife.43551.017

The following source data and figure supplement are available for figure 5:

**Source data 1.** This spreadsheet contains the behavioral responses for individual rats in *Figure 5*.
DOI: https://doi.org/10.7554/eLife.43551.019
**Figure supplement 1.** Data from the reinforced phase of reward devaluation testing for rats expressing the inhibitory DREADD hM4Di or mCherry following vehicle or CNO treatment in Experiment 4.
DOI: https://doi.org/10.7554/eLife.43551.018

increased reward seeking, generally, but also increased the likelihood that rats would attempt to retrieve reward after performing such actions. These behaviors were differentially mediated by the mesolimbic dopamine system. Specifically, chemogenetic inhibition of VTA dopamine neurons or their inputs to NAc, but not mPFC, disrupted the excitatory influence of the CS+ on reward seeking, but spared that cue's ability to increase attempts to retrieve reward. These behaviors were also differentially sensitive to reward devaluation, which suppressed reward seeking but actually increased the likelihood that rats would attempt to retrieve reward. VTA dopamine neurons inhibition did not impact the influence of reward devaluation on either component of behavior.

We found that attempts to retrieve reward by transitioning from the lever to the food cup were executed in a habitual manner, without consideration of reward value, consistent with action chunking (*Dezfouli et al., 2014*; *Smith and Graybiel, 2016*). However, task performance was not limited to these press-approach action chunks. When rats pressed the lever but were not reinforced (with food or cues), they would occasionally check the food cup but often omitted this response. This sporadic pattern of reward retrieval is adaptive given that strict press-approach action sequencing is unnecessary under such conditions, when rewards are sparse and uncertain. Instead, rats seemed to vacillate between two different strategies when initiating the lever-press response, performing it as part of a complete action chunk (press-approach) or as a discrete action (press only). These distinct patterns of reward seeking appeared to be differentially sensitive to reward devaluation. While rats were generally less likely to lever press for the devalued reward than for the valued reward, press-approach action chunks tended to be less sensitive to reward devaluation than presses that were not followed by approach. Because of this differential sensitivity to reward devaluation, the *proportion* of all lever presses followed by an attempt to retrieve reward was actually *greater* for devalued action than for the valued action. Such findings supports the connection between action chunking and habitual behavior (*Graybiel, 2008*; *Dezfouli et al., 2014*; *Smith and Graybiel, 2016*), and suggest that moment-to-moment control over self-paced, reward-seeking behavior may shift back and forth between habit and goal-directed systems.

PIT testing revealed that the CS+ generally increased lever pressing, but disproportionately increased the performance of press-approach action chunks, at least relative to their otherwise low frequency of occurring in the absence of the CS+. This finding further bolsters the connection between action chunking and habitual control given previous reports that habitual reward-seeking actions are particularly sensitive to the motivational effects of reward-paired cues (*Holland, 2004*; *Wiltgen et al., 2012*). However, while press-approach action chunks were elevated during the CS+, they still accounted for only a minority (between 30% and 50%) of lever presses that were performed during these trials. Most lever presses evoked by the CS+ were *not* followed by a food-cup approach, and it was this component of the PIT effect that was selectively disrupted by chemogenetic inhibition of VTA dopamine neurons or their inputs to NAc. The ability of the CS+ to promote press-approach chunks was, in contrast, completely spared by these manipulations. Consistent with this, we found that the response-suppressive effect of NAc dopamine terminal inhibition varied across rats based on the way they normally responded to the CS+. Rats that responded to that cue with a large increase in discrete lever presses (i.e., *without* subsequent food-cup approach) showed the greatest suppression. We suggest that this may reflect differences across rats in their sensitivity to the dopamine-dependent motivational effects of reward-paired cues.

Previous studies have found that dopamine receptor antagonists either selectively suppress lever pressing without affecting concomitant food-cup approach (*Nelson and Killcross, 2013*), or suppress both types behavior to a similar extent (*Wassum et al., 2011*; *Ostlund et al., 2012*). Even this latter finding is consistent with dopamine contributing more to reward seeking than reward retrieval, since a reduction in reward seeking creates fewer opportunities to retrieve reward. Interpreting these findings is problematic, however, because such studies typically have not applied microstructural analyses, like those used here, to distinguish between press-contingent and noncontingent food-cup approaches. One exception is a study by *Nicola (2010)* showing that blocking dopamine receptors in the NAc attenuates cue-triggered lever pressing without impacting the latency of subsequent food-cup approach behavior. Building on such findings, the current study used the PIT paradigm to show that the mesolimbic dopamine system specifically mediates the motivational influence of reward-paired cues on reward seeking but not their dissociable ability to increase the likelihood that such actions will be followed by an attempt to retrieve reward.

Our previous studies monitoring mesolimbic dopamine release during PIT performance are also interesting to consider together with the current findings. For instance, we found that CS+ evoked phasic dopamine release in the NAc correlates with that cue's effect on lever pressing (*Wassum et al., 2013*; *Ostlund et al., 2014*) but not food-cup approaches (*Aitken et al., 2016*). We also found that individual CS+ evoked lever presses are temporally correlated with transient bouts of phasic dopamine release (*Ostlund et al., 2014*). The current findings suggest that this relationship between NAc dopamine release and cue-motivated reward seeking may be stronger for discrete presses that are performed without a subsequent food-cup approach than for complete press-approach chunks. This question remains to be investigated, and would help resolve whether the

mesolimbic dopamine system is involved in modulating reward seeking, generally, or whether its activity becomes uncoupled from the execution of action chunks, which may become differentially associated with nigrostriatal dopamine system activity (*Jin and Costa, 2010*).

While dopamine is known to play a crucial role in forming new action chunks (*Graybiel, 1998*; *Jin and Costa, 2015*), its role in the expression of previously learned action chunks is less clear. Our findings indicate that VTA dopamine circuitry does not play a necessary role in the execution of press-approach action chunks, regardless of whether they are self-initiated or are prompted by a reward-paired cue. This is generally compatible with previous findings. For instance, dopamine receptor blockade suppresses action sequence performance early but not late in training (*Levesque et al., 2007*; *Wassum et al., 2012*). Moreover, the phasic NAc dopamine release that normally precedes action sequence performance tends to become attenuated as rats acquire efficient task performance, presumably through action chunking (*Cacciapaglia et al., 2012*; *Wassum et al., 2012*; *Klanker et al., 2015*; *Collins et al., 2016*). That said, the mesolimbic dopamine system continues to contribute to action sequence tasks that require considerable effort, such as the execution of a long series of lever presses (*Fischbach-Weiss et al., 2018*).

Inhibiting VTA dopamine neurons did not impact rats' sensitivity to reward devaluation, which is consistent with other findings in the literature (*Dickinson et al., 2000*; *Lex and Hauber, 2010a*; *Lex and Hauber, 2010b*; *Wassum et al., 2011*). Such findings are interesting given that regions innervated by this dopamine system, including the NAc and mPFC, are known to make important contributions to goal-directed decision making (*Bradfield and Balleine, 2017*; *Sharpe et al., 2019*). Of course, dopamine likely contributes to goal-directed decision making in more demanding tasks that require greater cognitive resources (*Floresco, 2013*; *Cools, 2015*; *Westbrook and Braver, 2016*).

It is also notable that inhibiting mPFC dopamine terminals had no detectable effects on expression of PIT, since food-paired cues are known to elicit dopamine release (*Bassareo and Di Chiara, 1997*; *Feenstra et al., 1999*) and neural activity (*Homayoun and Moghaddam, 2009*) in the mPFC. It is possible that the dissociable effects of NAc versus mPFC dopamine terminal inhibition reported here may relate to inherent differences between the mesolimbic and mesocortical dopamine systems, which include regional differences in release kinetics and in the density of dopamine terminals or receptors (*Lammel et al., 2008*; *Weele et al., 2019*; *Mahler et al., 2019*). However, previous lesion studies suggest that the mPFC may not be an essential component of the circuitry that mediates PIT performance (*Cardinal et al., 2003*; *Corbit and Balleine, 2003*), which is more in line with the current results.

Our findings may also have implications for understanding the role of dopamine in pathologies of behavioral control such as obsessive-compulsive disorder (OCD). In the signal attenuation model of OCD (*Joel and Avisar, 2001*), rats learn that response-contingent cues no longer signal that an instrumental reward-seeking action will produce reward. In this case, the logical organization of reward-seeking and -retrieval actions disintegrates, such that rats exhibit persistent reward seeking, typically without attempting to collect reward from the food cup. It was previously reported that blocking D1-dopamine receptors disrupts expression of these incomplete bouts of compulsive-like reward seeking, without affecting the production of complete bouts of reward seeking and retrieval, which continue to be performed on some test trials (*Joel and Doljansky, 2003*). Considered in this light, our findings suggest that the mesolimbic dopamine system may mediate the tendency for reward-paired cues to promote this potentially compulsive component of cue-motivated reward seeking. This link deserves further research, and may facilitate research to advance understanding and treatment of compulsive disorders like OCD and addiction (*Joel et al., 2008*; *Robinson et al., 2014*).

## Materials and methods

### Animals

In total, 89 male and female Long-Evans Tyrosine hydroxylase (Th):Cre+ rats (hemizygous Cre+) (*Witten et al., 2011*; *Mahler et al., 2019*) and wildtype (WT) littermates were used for this study. Subjects were at least 3 months of age at the start of the experiment and were single- or paired-housed in standard Plexiglas cages on a 12 hr/12 hr light/dark cycle. Animals were maintained

at ~85% of their free-feeding weight during behavioral procedures. All experimental procedures that involved rats were approved by the UC Irvine Institutional Animal Care and Use Committee and were in accordance with the National Research Council Guide for the Care and Use of Laboratory Animals.

## Apparatus

Behavioral procedures took place in sound- and light-attenuated Med Associates chambers (St Albans, VT, USA; ENV-007). Individual chambers were equipped with two retractable levers (Med Associates; ENV-112CM) positioned to the left and right of recessed food cup. Grain-based dustless precision pellets (45 mg, BioServ, Frenchtown, NJ, USA) were delivered into the cup using a pellet dispenser (Med Associates; ENV-203M-45). Sucrose solution (20% wt/vol) was delivered into the cup with a syringe pump (Med Associates; PHM-100). A photobeam detector (Med Associates; ENV-254-CB) positioned across the magazine entrance was used to record food-cup approaches. Chambers were illuminated by a houselight during all sessions.

## Surgery

Th:Cre+ rats were anesthetized using isoflurane and placed in a stereotaxic frame for microinjections of a Cre-dependent (DIO) serotype two adeno-associated virus (AAV) vectors to induce dopamine neuron-specific expression of the inhibitory designer receptor exclusively activated by designer drug (DREADD) hM4Di fused to mCherry (AAV-hSyn-DIO-hM4Di-mCherry), or mCherry alone (AAV-hSyn-DIO-mCherry) (University of North Carolina Chapel Hill vector Core, Chapel Hill, NC, USA/Addgene, Cambridge, MA, USA; Experiment 2 was replicated with both sources) (*Armbruster et al., 2007*; *Mahler et al., 2019*). The AAV was injected bilaterally into the VTA (−5.5 mm AP,±0.8 mm ML, −8.15 mm DV; 1μL/side). Experiment 3 rats were bilaterally implanted with guide cannulae (22 gage, Plastic One) 1 mm dorsal to NAc (+1.3 AP, ±1.8 ML, −6.2 DV) or mPFC (+3.00 AP, ±0.5 ML, −3.0 DV) for subsequent clozapine-n-oxide (CNO) microinjections. Animals were randomly assigned to virus (hM4Di or mCherry) and cannula location (NAc or mPFC) groups. Animals were allowed at least 5 days of recovery before undergoing food restriction and behavioral training. Testing occurred at least 25 days after surgery to allow adequate time for viral expression of hM4Di throughout dopamine neurons, including in terminals within the NAc and mPFC.

## Experiment 1: Effects of response-contingent feedback about reward delivery on reward retrieval

### Instrumental learning

WT rats (n = 9) underwent 2 d of magazine training. In each session, 40 pellets were delivered into the food cup on a random 90 s intertrial interval (ITI). Rats then received 9 d of instrumental lever-press training. In each session, rats had continuous access to the right lever, which could be pressed to deliver food pellets into the food cup. The schedule of reinforcement was adjusted over days from continuous reinforcement (CRF) to increasing random intervals (RI), such that reinforcement only became available once a randomly determined interval had elapsed since the last reinforcer delivery. Rats received one day each of CRF, RI-15s, and RI-30s training, before undergoing 6 days of training with RI-60s. Each session was terminated after 30 min or after 20 rewards deliveries.

### Varying response-contingent feedback

Following training, rats were given a series of tests to assess the influence of response-contingent feedback about reward delivery on instrumental reward-seeking (lever presses) and reward-retrieval responses (press-contingent food-cup approach). Rats were given three tests (30 min each, pseudo-random order over days) during which lever pressing caused: 1) activation of the pellet dispenser to deliver a pellet into the food cup (RI-60s schedule; *Food and Cues Test*), 2) activation of the pellet dispenser to deliver a pellet into an external cup not accessible to the rats, producing associated sound and tactile cues but no reward (also RI-60s schedule; *Cues Only Test*), or 3) no dispenser activation (i.e., extinction; *No Food or Cues Test*).

## Experiments 2 and 3: Role of mesocorticolimbic dopamine in cue-motivated reward seeking and retrieval

### Pavlovian conditioning

Th:Cre+ rats (n = 60) underwent 2 d of magazine training, as in Experiment 1 (40 pellets on 90 s random ITI). Rats then received eight daily Pavlovian conditioning sessions. Each session consisted of a series of 6 presentations of a two-min audio cue (CS+; either a pulsating 2 kHz pure tone (0.1 s on and 0.1 s off) or white noise; 80 dB), with trials separated by a 5 min variable ITI (range 4–6 min between CS onsets). During each CS+ trial, pellets were delivered on a 30 s random time schedule, resulting in an average of 4 pellets per trial. Rats were separately habituated to an unpaired auditory stimulus (CS-; alternative audio stimulus; 2 min duration). CS- exposure procedures differed slightly across experiments. For Experiment 2, which assessed the effects of system-wide dopamine neurons inhibition, rats received a final Pavlovian conditioning session consisting of four trials with the CS + (reinforced, as described above) followed by four trials with the CS- (nonreinforced), separated by a 5 min variable ITI. In Experiment 3, which assessed the effects of local inhibition of dopamine terminals in NAc or mPFC, rats were given 2 days of CS- only exposure (eight nonreinforced trials per session, 5 min variable ITI) following initial CS+ training. Conditioning was measured by comparing the rate of food-cup approach between the CS onset and the first pellet delivery (to exclude unconditioned behavior) to the rate of approach during the pre-CS period.

### Instrumental training

Following Pavlovian conditioning, rats were given 9 d of instrumental training, as in Experiment 1, with one day each of CRF, RI-15s, RI-30s, and 6 days of RI-60s. Sessions ended after 30 min or 20 rewards were earned.

### Pavlovian-to-instrumental transfer (PIT) test

After the last instrumental training session, rats were given a session of Pavlovian (CS+) training, identical to initial training. They were then given a 30 min extinction session, during which lever presses were recorded but had no consequence (i.e., no food or cues). On the next day, rats were given a PIT test, during which the lever was continuously available but produced no rewards. Following 8 min of extinction, the CS+ and CS- were each presented four times (2 min per trial) in pseudo-random order and separated by a 3 min fixed ITI. Before each new round of testing, rats were given two sessions of instrumental retraining (RI-60s), one session of CS+ retraining, and one 30 min extinction session, as described above. Test procedures differed slightly between Experiments 2 and 3.

### Experiment 2

Th:Cre+ rats expressing hM4Di (n = 18) or mCherry only (n = 14) in VTA dopamine neurons were used to assess the effects of system-wide inhibition of the mesocorticolimbic dopamine system on PIT performance. These groups were run together and received CNO (5 mg/kg, i.p.) or vehicle (5% DMSO in saline) injections 30 min prior to testing. They underwent a second test following retraining (described above), prior to which the alternative drug pretreatment was administered.

### Experiment 3

In Experiment 3A, Th:Cre+ rats expressing hM4Di in VTA dopamine neurons were used to assess the impact of locally inhibiting dopaminergic terminals in the NAc (n = 7) or mPFC (n = 9) on PIT performance. Because microinjection procedures produced additional variability in task performance, rats in this experiment underwent a total of 4 tests. Rats received either CNO microinfusions (1 mM, 0.5 μL/side or 0.3 μL/side, for NAc and mPFC respectively) or vehicle (DMSO 5% in aCSF) 5 min before the start of each test and were given two rounds of testing each with CNO and vehicle (test order counterbalanced across other experimental conditions). To determine if the effects of CNO microinjections depended on hM4Di expression, a separate control study (Experiment 3B) was run using Th:Cre+ rats expressing mCherry only in VTA dopamine neurons. Experiments 3A and 3B were run and analyzed separately.

## Experiment 4: Role of mesocorticolimbic dopamine in goal-directed action selection

### Instrumental Training

Th:Cre+ rats expressing hM4Di (n = 11) or mCherry only (n = 9) in VTA dopamine neurons began with 2 d of magazine training, during which they received 20 grain-pellets and 20 liquid sucrose rewards (0.1 mL of 20% sucrose solution, wt/vol) in random order according to a common 30 s random ITI. This was followed by 11 d of instrumental training with two distinct action–outcome contingencies (e.g., left-lever press → grain; right-lever press→ sucrose). The reinforcement schedule that was gradually shifted over days with 2d of CRF to increasingly effortful random ratio (RR) schedules, with 3 d of RR-5, 3 d of RR-10, and 3d of RR-20 reinforcement. The left and right lever-press responses were trained in separate sessions, at least 2 hr apart, on each day. Action-outcome contingencies were counterbalanced across subjects. Sessions were terminated after 30 min elapsed or 20 pellets were earned.

### Devaluation Testing

To selectively devalue one of the food rewards prior to testing, rats were satiated on grain pellets or sucrose solution by providing them with 90 min of unrestricted access to that food in the home cage. After 60 min of feeding, rats received CNO (5 mg/kg, i.p.) or vehicle injections. After an additional 30 min of feeding, rats were placed in the chamber for a test in which they had continuous access to both levers. The test began with a 5 min nonreinforced phase (no food or cues), which was immediately followed by a 15 min reinforced phase, during which each action was reinforced with its respective reward (CRF for the first five rewards, then RR-20 for the remainder of the session). Rats were given a total of 4 devaluation tests, two after CNO and two after vehicle, alternating the identity of the devalued reward across the two tests in each drug condition (test order counterbalanced across training and drug conditions).

### Histology

Rats were deeply anesthetized with a lethal dose of pentobarbital and perfused with 1x PBS followed by 4% paraformaldehyde. Brains were postfixed in 4% paraformaldehyde, cryoprotected in 20% sucrose and sliced at 40 μm on a cryostat. To visualize hM4Di expression, we performed immunohistochemistry for Th and mCherry tag. Tissue was first incubated in 3% normal donkey serum PBS plus Triton X-100 (PBST; 2 hr) and then in primary antibodies in PBST at 4°C for 48 hr using rabbit anti-DsRed (mCherry tag; 1:500; Clontech; 632496), and mouse anti-Th (1:1,000, Immunostar; 22941) antibodies. Sections were incubated for 4 hr at room temperature in fluorescent conjugated secondary antibodies (Alexa Fluor 488 goat anti-mouse (Th; 1:500; Invitrogen; A10667) and Alexa Fluor 594 goat anti-rabbit (DsRed; 1:500; Invitrogen; A11037)).

### Drugs

CNO was obtained from NIMH (Experiments 2 and 4) or Sigma-Aldrich (St. Louis, MO, USA; Experiment 3), and dissolved in 5% DMSO in saline, or aCSF for microinjection.

## Behavioral measures

Reward-*seeking* actions were quantified as the total number (frequency) of lever presses performed per unit time. Based on microstructural analyses described below, lever presses that were followed by a food-cup approach (≤2.5 s) were distinguished from presses that were not followed by an approach. The proportion of presses that were followed by an approach response served as our primary measure of press-contingent reward retrieval. We also analyzed bouts of noncontingent food-cup approach (occurring >2.5 s after the most recent press or approach), which served as a measure of spontaneous or cue-evoked reward retrieval.

## Statistical analysis

Data were analyzed using general(ized) linear mixed-effects models (*Pinheiro and Bates, 2000*), which allows for simultaneous parameter estimation as a function of condition (fixed effects) and the individual rat (random effects) (*Pinheiro and Bates, 2000*; *Bolker et al., 2009*; *Boisgontier and Cheval, 2016*). Analyses on count data (e.g., response frequency) incorporated a Poisson response

distribution and a log link function (*Coxe et al., 2009*). Fixed-effects structures included an overall intercept and the full factorial of all primary manipulations (Experiment 2: Group, Drug, CS Type, CS Period; Experiment 3: Site, Drug, CS Type, CS Period; Experiment 4: Group, Drug, Lever), and the random-effects structures included by-subjects uncorrelated intercepts adjusted for the within-subjects manipulations (i.e., Experiments 2 and 3: Drug, CS Type, and CS Period; Experiment 4: Drug, Lever). 'CS Type' refers to the distinction between the CS+ and CS-, while 'CS Period' refers to the distinction between the 120 s CS duration and the 120 s period preceding its onset. Proportion data were square-root transformed prior to analysis to correct positive skew, but are plotted in non-transformed space for ease of interpretation. These data were collapsed across pre-CS+ and pre-CS-periods, such that the factor 'CS Period' had three levels (CS+, CS-, and Pre-CS). The fixed- and random-effects structures of this analysis was identical to the frequency analysis above with the exception that CS Type was not included in the analysis, and the random-effects structure only included by-subjects intercepts.

All statistical analyses were conducted using the Statistics and Machine Learning Toolbox in MATLAB (The MathWorks; Natick, MA, USA). The alpha level for all tests was .05. As all predictors were categorical in the mixed-effects analysis, effect size was represented by the unstandardized regression coefficient (*Baguley, 2009*), reported as *b* in model output tables. Mixed-effects models provide *t*-values to reflect the statistical significance of the coefficient relative to the population mean (i.e., simple effects). These simple effects are indicative of main effects and interactions when a factor has only two levels. For factors with at least three levels, *F*-tests were conducted to reveal the overall significance of the effect or interaction(s) involving this factor. The source of significant interactions was determined by secondary mixed-effects models identical to those described above but split by the relevant factor of interest. For analyses in which a significant main effect had more than two levels, post-hoc tests of main effects employed MATLAB's *coefTest* function, and interactions were reported in-text as the results of ANOVA *F*-tests (i.e., whether the coefficients for each fixed effect were significantly different from 0).

When analyzing data from PIT experiments, the ability of the CS+ to selectively increase performance of a response (relative to the CS-) over baseline (pre-CS) levels was indicated by a significant CS Type * CS Period interaction. We were particularly interested in treatment-induced alterations in the expression of this effect, as indicated by significant 3-way and 4-way interactions involving this CS Type * CS Period term, in combination with Drug and/or Group factors. We were also interested in potential main effects of Drug and/or Group factors, reflecting broad, cue-independent behavioral effects. While statistical output tables include a summary of all fixed effects included in the model, only these theoretically interesting findings are discussed in the main text. Lower level interactions involving only CS Type *or* CS Period, but not their combination, are provided in the output tables but are not discussed in the main text given that they may be the product of incidental or spurious behavioral differences across cue conditions.

PIT Scores (CS+ – pre-CS+) were calculated for more focused analysis of CS+ elicited lever pressing. One-sample *t*-tests were used to assess the effect of CNO for each group. Because inhibiting VTA dopamine neurons or their NAc terminals predominantly disrupted the ability of the CS+ to elicit lever presses that were not followed by an approach response, we also assessed if differences across rats in their tendency to exhibit such behavior in the Vehicle Test (PIT score; presses without approach) correlated with differences in their sensitivity to the response-suppressive effect of CNO on CS+ elicited lever pressing (CNO – Vehicle; PIT score, all presses).

## Acknowledgements

The authors acknowledge the assistance of Christy N Munson in the acquisition of behavioral data.

## Additional information

### Competing interests

Kate M Wassum: Reviewing editor, *eLife*. The other authors declare that no competing interests exist.

## Funding

| Funder | Grant reference number | Author |
|---|---|---|
| National Institute on Drug Abuse | | Stephen V Mahler |
| National Institute of Mental Health | 106972 | Kate M Wassum<br>Sean B Ostlund |
| National Institute of Diabetes and Digestive and Kidney Diseases | 098709 | Sean B Ostlund |
| National Institute on Drug Abuse | 029035 | Sean B Ostlund |
| National Institute on Aging | 045380 | Sean B Ostlund |

The funders had no role in study design, data collection and interpretation, or the decision to submit the work for publication.

## Author contributions

Briac Halbout, Andrew T Marshall, Conceptualization, Data curation, Formal analysis, Investigation, Visualization, Methodology, Writing—original draft, Writing—review and editing; Ali Azimi, Data curation, Investigation; Mimi Liljeholm, Conceptualization, Methodology, Writing—review and editing; Stephen V Mahler, Conceptualization, Validation, Investigation, Methodology, Writing—review and editing; Kate M Wassum, Conceptualization, Supervision, Funding acquisition, Writing—review and editing; Sean B Ostlund, Conceptualization, Data curation, Formal analysis, Supervision, Funding acquisition, Investigation, Methodology, Writing—original draft, Project administration, Writing—review and editing

## Author ORCIDs

Briac Halbout (ID) https://orcid.org/0000-0001-6128-2601
Andrew T Marshall (ID) https://orcid.org/0000-0002-0068-8138
Mimi Liljeholm (ID) http://orcid.org/0000-0001-9066-6989
Stephen V Mahler (ID) https://orcid.org/0000-0002-8698-0905
Sean B Ostlund (ID) https://orcid.org/0000-0003-1635-3911

## Ethics

Animal experimentation: All experimental procedures that involved rats were approved by the UC Irvine Institutional Animal Care and Use Committee (protocol AUP-17-68) and were in accordance with the National Research Council Guide for the Care and Use of Laboratory Animals.

## Decision letter and Author response

Decision letter https://doi.org/10.7554/eLife.43551.023
Author response https://doi.org/10.7554/eLife.43551.024

# Additional files

## Supplementary files

• Supplementary file 1. Generalized linear mixed-effects model outputs.
DOI: https://doi.org/10.7554/eLife.43551.020

• Transparent reporting form
DOI: https://doi.org/10.7554/eLife.43551.021

## Data availability

All data generated and analyzed during this study are included in supporting files. Source data files have been provided for Figures 1, 3, 4 and 5, as well as their respective figure supplements.

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
