## [Decision Letter]

[Editors’ note: this article was originally rejected after discussions between the reviewers, but the authors were invited to resubmit after an appeal against the decision.]

Thank you for submitting your work entitled "Ventral tegmental dopamine inputs to the nucleus accumbens mediates cue-triggered motivation but not reward expectancy" for consideration by *eLife*. Your article has been reviewed by three peer reviewers, and the evaluation has been overseen by a Reviewing Editor and a Senior Editor. The following individual involved in review of your submission has agreed to reveal their identity: Stan Floresco (Reviewer #3).

Our decision has been reached after consultation between the reviewers. Based on these discussions and the individual reviews below, we regret to inform you that your work will not be considered further for publication in *eLife*, at least in its current form.

Halbout and colleagues performed a series of experiments to determine the role of VTA dopamine neurons projecting to the nucleus accumbens (NAc) in reward-seeking behaviors in a Pavlovian-instrumental transfer (PIT) task. Specifically, the authors sought to separate whether dopamine regulates reward expectancy or motivation (response vigor). The authors show that chemogenetic inhibition of dopamine neurons impaired CS-induced lever pressing but not entry into the reward port (reward retrieval) induced directly by the sensory stimuli associated with food delivery. Based on the idea that the reward retrieval is caused by reward expectancy whereas CS-induced lever pressing is due to motivation, the authors conclude that VTA dopamine neurons regulate motivation (response vigor) but not reward expectancy.

All the reviewers thought that this study addresses an important question: identifying the exact role that this population of dopamine neurons plays in reward-seeking behavior. They agreed that this study uses very careful behavioral observations and a nice combination of behavioral paradigms, and obtained important data. However, the overall assessment of the study somewhat varied among the reviewers, as you can see below. After discussion, we thought that the definitions of some terms such as 'reward expectancy' are not very clear. 'Reward expectancy' is rather a broad term and has been used differently in different fields. It is not obvious whether 'motivation' and 'reward expectancy' are completely separated, in the first place. Also, reward expectancy may, in some literature, imply a narrower definition corresponding to that required for goal-directed behaviors. Overall, the definition of reward expectancy based on reward retrieval behavior is not very convincing (please see the comment by reviewer 1 and 2) and some of the results (devaluation experiments) appear to contradict the idea of 'reward expectancy'. Furthermore, as reviewer 1 pointed out, there appear to be alternative explanations of the data. In total, we thought that these terminologies and the interpretation of the results require significant revisions for clarify and consistency. Please also note that *eLife* is a general bio logy journal, and writing has to be understandable to readers outside a subfield. Additionally, the reviewers were concerned by the relatively small effects. Specifically, in some experiments (Experiment 2 [Figure 3B] and Experiment 3 [Figure 4B and Figure 4—figure supplement 2]), it appears that CNO alone had a strong effect on behavior and the results using a summary statistics ('CNO suppression') are not very convincing.

Given the above concerns, we cannot proceed with the current manuscript. Due to the potential importance of the data, however, we would like to suggest the authors to submit a revision plan if the authors think that they can address the major concerns raised by the reviewers. The revision plan should respond to the reviewers' main criticisms and describe specifically how the paper would be changed to address them, including whether new experiments would be performed. We would like to use the revision plan to make a final decision as to whether a revision is a viable approach for this manuscript.

Reviewer #1:

Halbout et al. use fine-grained analysis of behavior in a Pavlovian-instrumental transfer (PIT) task to assay whether VTA dopamine neurons facilitate the general PIT effect via increasing reward expectancy (as opposed to enhancing cue-triggered motivation). This is an important question, as answering it would help to further define dopamine function. The authors' strategy is to use the probability of reward port entry after an operant response as an index of reward expectancy. This measure is not reduced by chemogenetic inhibition of VTA dopamine neurons even though this manipulation effectively reduces the main PIT effect of enhanced operant responding during presentation of Pavlovian cues. The authors conclude that dopamine does not enhance PIT via promoting reward expectancy. Instead, dopamine enhances PIT via its contribution to cue-triggered motivation, and an additional non-dopamine-dependent component of PIT is the result of enhanced reward expectancy due CS+ presentation.

Although for the most part the results are convincing from the technical perspective, the interpretation completely depends on the authors' claim that port entries occurring shortly after an operant response are an index of reward expectancy. The authors support this claim by showing, in an instrumental task identical to the one used in the PIT experiments, that cues presented after the operant response that signal reward delivery (pellet drop) increase the probability of a port entry. However, this unsurprising result is entirely consistent with an alternative interpretation that the authors have apparently not considered. That is that port entry is simply the next step after the lever response in a chunked skilled action sequence. Such sequences are most likely a form of stimulus-response habit in which the stimulus is the previous action. In addition, because pellet drop occurs after the lever press and predicts reward, it could also participate in S-R habit learning. In this scenario, a lever press followed by pellet drop would simply activate the response (port entry) more strongly than either stimulus alone. Supporting this interpretation, the authors show that port entry after lever pressing is strongly resistant to outcome devaluation – a hallmark of S-R habits. It is therefore perplexing that the authors use this behavior as an index of reward expectancy – a component of goal-directed and cognitive forms of behavior, which should be sensitive to devaluation precisely because the value of the expected reward is reduced.

The authors go on to show that during the PIT test, not only does the CS+ increase the probability of the operant response relative to control periods (e.g., CS-), but it also increases the probability that any individual operant response will be followed by a port entry. They further show that chemogenetic inhibition of VTA dopamine neurons or terminals in the accumbens during the PIT test reduces the CS-induced increase in operant responding while leaving intact the probability of port entry after an operant response. The authors conclude that although a CS-triggered increase in reward expectancy drives a component of the PIT effect (those operant responses followed by port entry), an additional component is not driven by expectancy (operant responses not followed by port entry), and dopamine contributes to PIT via the latter mechanism, likely via enhancing cue-triggered motivation.

To accept this interpretation, one must accept the assertion that operant-entry sequences are the result of reward expectancy to a greater extent than operant responses occurring without a subsequent entry. As described above, this claim is suspect. It is certainly interesting that inhibition of dopamine neurons has a greater effect on single operants than operant-entry sequences, but there are alternative explanations. For instance, the authors' devaluation experiment suggests that operant actions that are not followed by port entry are sensitive to devaluation and are therefore likely controlled by action-outcome goal-directed associations. In contrast, the post-devaluation increase in frequency of operant-entry sequences suggests that operant-entry behavior is at least partially controlled by stimulus-response habit associations. In this scenario, dopamine facilitates the influence of a Pavlovian CS+ on goal-directed behavior but not habit behavior. Under the assumption that reward expectancy is more important for goal-directed than habit behaviors, this is roughly the opposite of the authors' conclusion. What the authors may have found is something perhaps even more intriguing: that CSs in the PIT test can influence habit-based behaviors at least as strongly as goal-directed behaviors. Of course, one might expect the neural mechanism of such effects to be different (as an interaction between Pavlovian and action value representations is possible only with goal-directed behaviors), and it is therefore informative that dopamine contributes only to facilitation of goal-directed behaviors.

In summary, the authors have found something intriguing in the resistance of operant response-port entry sequences to inhibition of dopamine neurons and dopamine terminals in the accumbens. However, their interpretation is not adequately supported by their data, some of which in fact contradict their conclusions.

Reviewer #2:

This article by Halbout and colleagues describe a series of experiments aimed at clarifying the contribution of the dopaminergic innervation of the ventral striatum in the interaction between Pavlovian and instrumental responses in an appetitive context. The study is well designed and this question should be relevant for neuroscientists interested in the neural basis of decision-making and motivation.

I have, however, a series of issues with the manuscript:

The terminology is very confusing, and the theoretical frame is globally difficult to grasp. There are many concepts that are used to capture very simple behaviors, even in the Results section (reward seeking, exploration, reward expectancy, motivation, vigor etc.). Could the authors make a set of simple predictions based on a simple theoretical frame and describe how they decide to operationalize the questions? There are places in the text when it is the case, but I'm afraid that there is a drift in the issues at stake between the Abstract and the Discussion.

The description of the methods is too superficial. It might be sufficient for specialists in the field (conditioning in rats) and/or people who know the work of this team, but not for a more general audience. Typically, including the contingencies and the timing of the reward schedules in the different tasks would help.

I am having some trouble reconciling the conclusions of the authors and the data, as shown by the figure:

Experiment 1: Again, the text is difficult to read but it seems to me that there are two things going on here: the operant behavior of these animals in the task is very 'habitual' (as opposed to goal-directed), since the rate of lever pressing decreases very little when reward is omitted (Figure 1D). In line with that, presumably, animals almost only try to collect the reward when they get a direct evidence (cue) that it has been delivered. So reward does not drive action and actions do not predict reward, and animals are left with Pavlovian processes to anticipate the reward. Is there more to it than that? If yes, it would require precise quantitative predictions and measures to demonstrate it. I would avoid the word 'exploration', unless the authors can demonstrate that animals try to get the reward by exploring another potential way to get the reward (e.g. another lever). Here, action seems to be driven much more by compulsion/ habits than exploration.

Experiments 2 and 3:

Looking at Figure 3B (Experiment 2) and Figure 4B and Figure 4—figure supplement 2 (Experiment 3), CNO alone has a strong effect on behavior and the difference between mCherry and hM4Di animals is really small. Not even sure it is significant, on the scale of the experiment. For example, in Experiment 2, Figure 3D shows a greater effect of CNO on response to CS- than CS+. Again, when looking at all the bars, I don't understand the conclusions.

In Experiment 2, to test the prediction regarding the role of DA in PIT, they should test the interaction between 3 factors (CNO vs. Vehicle; mCherry vs. hM4Di and pre vs. post CS+). By the way, why using the pre/post CS+ when there is a CS- condition, which seems a priori as a better reference for assessing the specificity of the Pavlovian effects of the CS+, as opposed to general effect of having a stimulus? Looking at the data, CNO abolishes the increase in responding evoked by the CS+, and the major difference between mCherry and hM4Di is in the pre CS+ period… If the authors really want to take into trial to trial variability, they could use a relative measure (pre vs. post, on a trial by trial basis) but compare the effect of CS+ with the effect of CS- to assess PIT, and the influence of the various manipulations.

It is essentially the same thing for experiment 3: the effect of CNO (here in the NAc) vs. vehicle is much bigger than the difference between mCherry and hM4Di. Ok, injecting CNO in the MFC has no effect, but this is potentially because the concentration of DA receptor (targeted by the CNO…) is smaller there than in the ACC. Separate issue, but as I mentioned above, describing what is essentially compulsive/habitual lever pressing as 'exploratory seeking' does not help.

Given these, I am not sure that any of the conclusions regarding the role of DA really stands. Again, the experimental design as a whole is excellent, I believe, and the results are interesting in terms of clarifying the neural and behavioral processes at stake in PIT, but ample revision are needed.

Reviewer #3:

This is a very interesting and cleverly-designed study that parses our how mesocorticolimbic dopamine (DA) transmission may be involved in the ability of Pavlovian cues to invigorate reward seeking, using a well-established Pavlovian-to-Instrumental transfer test. Using DREADD approaches to silence dopamine cell bodies they show that suppressing DA activity blocks or attenuates the PIT effect. Moreover, this appeared to be mediated primarily by DA activity in the accumbens, but not prefrontal cortex. What was particularly important about these findings is that the authors conducted a sophisticated microstructural analysis of behavior, focusing on how often rats may approach the food cup (as a measure of reward expectancy). This analysis showed that, even though reducing DA activity suppressed the increase in lever pressing induced by Pavlovian, reward-associated cues, it did not affect approaches to the food cup, suggesting that DA's role in this context is to enhance motivation, but not reward expectancy. They also showed that reducing DA activity did not influence how reinforcer devaluation altered responding.

This is a well-designed study that has important implications for teasing out what DA does (and does not do) in modulating reward seeking. The analyses are sound, and the Discussion is fair and scholarly. I have no major issues with the paper, but a few relatively minor points the authors should attend to.

1) Subsection “Pavlovian conditioning”, was a 10 Hz tone used? Or 10 kHz (I'm not sure rats can hear 10 Hz).

2) "After a session of Pavlovian conditioning, rats were given a 30-min extinction session" – based on the preceding section, it's unclear what happened. Did rats receive a Pavlovian session after the instrumental training and before the first test and then every subsequent one? This should be clarified.

3) Subsection “Inhibiting dopamine neurons during Pavlovian-to-instrumental transfer preferentially disrupts cue-motivated reward seeking, but not reward retrieval”, last paragraph – the reader is referred to Figure 3E and F, but there is no F panel. Do they mean D and E?

4) In looking at the mCherry group in Figure 3, it appears that systemic CNO may have attenuated the PIT effect (although clearly not as much as it did in the hM4Di group). I'm not sure if the analysis look at this comparison, but it would be important to point out either way if CNO on its own has some effect on PIT.

5) The authors use an interesting correlational analysis to try explain the variation in the accumbens CNO data. However, looking at the histology, there is also considerable variation in the placements of infusions, some in core, some in shell. This warrants some mention, and perhaps another analysis seeing if the effects were stronger in animals with placements located in one NAc subregion vs. another.

[Editors’ note: what now follows is the decision letter after the authors submitted for further consideration.]

Thank you for resubmitting your work entitled "Mesolimbic dopamine projections mediate cue-motivated reward seeking but not reward retrieval" for further consideration at *eLife*. Your revised article has been favorably evaluated by Joshua Gold as the Senior Editor, a Reviewing Editor, and three reviewers.

All of the reviewers agreed that the manuscript has been greatly improved. However. the reviewer 1 still has remaining issues as described below. Before proceeding further, we would like to see your response to these comments.

Reviewer #1:

Halbout et al. show that inhibition of dopamine neurons (or terminals in the nucleus accumbens, but not prefrontal cortex) reduces the general Pavlovian-instrumental transfer (PIT) effect primarily by reducing operant responses that occur without a following reward port entry. In rats trained to perform an instrumental task, reward devaluation reduced operant performance in an extinction test, but left operant response-port entry sequences relatively unaffected, indicating that performance of operant-entry sequences is most likely a stimulus-response habit. These results make the subtle but important point that even though cues that motivate reward seeking increase the number of action sequences performed via a mesolimbic dopamine-dependent mechanism, the performance of the sequence once it is initiated is independent of mesolimbic dopamine. This point is in line with previous work from the authors and others, and its support with microstructural behavioral analysis provides a careful and useful addition to the field.

This version of the paper includes additional control experiments, and the authors' new interpretations are much better supported by their data than previously. However, the authors should address a few additional points.

1) The authors present evidence that the chunked operant-entry action sequence is resistant to devaluation. However, it is possible that port entry behavior is simply resistant to devaluation whether or not the entry follows an operant response. The authors show that animals make "noncontingent food cup approaches" (i.e., port entries), but do not analyze the rate of this behavior in any of their tasks. The argument that operant responses are resistant to devaluation because they are part of a chunked, habit action sequence would be strengthened if entries that are NOT part of such sequences were not resistant to devaluation.

2) It would also be interesting to know whether noncontingent entries in the PIT test are dependent on dopamine/mesolimbic dopamine.

3) Systemic CNO had a greater effect on PIT than intra-accumbens CNO. The authors should discuss the possible reasons for this.

4) Figure 4F and the similar supplementary figure are hard to understand. First, the X axis should be more descriptive (i.e., refer to CS-induced increase in lever presses without approach). Second, the Y axis reads "CNO suppression", but apparently greater suppression is represented by more negative numbers. This is counterintuitive. I think the authors are trying to say that the impact of CNO positively correlates with the impact of the CS on lever presses without approach, but they write that there is a negative correlation. The figure legend adds to the confusion by saying that the Y axis is "PIT Score for vehicle test – PIT Score for CNO test". If that's the case, then the lower PIT scores in CNO should result in positive values, not the predominantly negative values shown in the figure.

Reviewer #2:

The authors seriously improved the manuscript and they addressed my previous remarks. This is a very interesting piece of work and I have no major concern anymore.

*Reviewer #3:*

In my opinion, the authors have done a fine job at addressing concerns raised during the previous rounds of review. I have no further comments. I reiterate that I believe this is an interesting and clever study.

---

## [Author Response]

[Editors’ note: the author responses to the first round of peer review follow.]

Reviewer #1:[…] Although for the most part the results are convincing from the technical perspective, the interpretation completely depends on the authors' claim that port entries occurring shortly after an operant response are an index of reward expectancy. The authors support this claim by showing, in an instrumental task identical to the one used in the PIT experiments, that cues presented after the operant response that signal reward delivery (pellet drop) increase the probability of a port entry. However, this unsurprising result is entirely consistent with an alternative interpretation that the authors have apparently not considered. That is that port entry is simply the next step after the lever response in a chunked skilled action sequence. Such sequences are most likely a form of stimulus-response habit in which the stimulus is the previous action. In addition, because pellet drop occurs after the lever press and predicts reward, it could also participate in S-R habit learning. In this scenario, a lever press followed by pellet drop would simply activate the response (port entry) more strongly than either stimulus alone. Supporting this interpretation, the authors show that port entry after lever pressing is strongly resistant to outcome devaluation – a hallmark of S-R habits. It is therefore perplexing that the authors use this behavior as an index of reward expectancy – a component of goal-directed and cognitive forms of behavior, which should be sensitive to devaluation precisely because the value of the expected reward is reduced.The authors go on to show that during the PIT test, not only does the CS+ increase the probability of the operant response relative to control periods (e.g., CS-), but it also increases the probability that any individual operant response will be followed by a port entry. They further show that chemogenetic inhibition of VTA dopamine neurons or terminals in the accumbens during the PIT test reduces the CS-induced increase in operant responding while leaving intact the probability of port entry after an operant response. The authors conclude that although a CS-triggered increase in reward expectancy drives a component of the PIT effect (those operant responses followed by port entry), an additional component is not driven by expectancy (operant responses not followed by port entry), and dopamine contributes to PIT via the latter mechanism, likely via enhancing cue-triggered motivation.To accept this interpretation, one must accept the assertion that operant-entry sequences are the result of reward expectancy to a greater extent than operant responses occurring without a subsequent entry. As described above, this claim is suspect. It is certainly interesting that inhibition of dopamine neurons has a greater effect on single operants than operant-entry sequences, but there are alternative explanations. For instance, the authors' devaluation experiment suggests that operant actions that are not followed by port entry are sensitive to devaluation and are therefore likely controlled by action-outcome goal-directed associations. In contrast, the post-devaluation increase in frequency of operant-entry sequences suggests that operant-entry behavior is at least partially controlled by stimulus-response habit associations. In this scenario, dopamine facilitates the influence of a Pavlovian CS+ on goal-directed behavior but not habit behavior. Under the assumption that reward expectancy is more important for goal-directed than habit behaviors, this is roughly the opposite of the authors' conclusion. What the authors may have found is something perhaps even more intriguing: that CSs in the PIT test can influence habit-based behaviors at least as strongly as goal-directed behaviors. Of course, one might expect the neural mechanism of such effects to be different (as an interaction between Pavlovian and action value representations is possible only with goal-directed behaviors), and it is therefore informative that dopamine contributes only to facilitation of goal-directed behaviors.In summary, the authors have found something intriguing in the resistance of operant response-port entry sequences to inhibition of dopamine neurons and dopamine terminals in the accumbens. However, their interpretation is not adequately supported by their data, some of which in fact contradict their conclusions.

We agree that our original focus on *reward expectancy* was confusing. This was partly due to our unconventional use of that term, which we did not intend to be synonymous with *goal-directed control* given evidence that reward expectancies can guide action selection in a manner that is independent of reward value during PIT performance (Colwill and Rescorla, 1990; Rescorla, 1994; Holland, 2004). But we now believe this framework for data interpretation was unclear and unnecessary. We have therefore omitted the term reward expectancy altogether, and have substantially revised the manuscript to focus on the broader topic of how dopamine contributes to the control of instrumental reward-seeking and reward-retrieval actions, while explicitly contrasting goal-directed (devaluation-sensitive) and habitual (devaluation-insensitive) accounts of behavior. From this perspective, we agree with the reviewer’s conclusion that press-contingent food-cup approaches were performed habitually, suggesting the use of a press-approach behavioral chain or action chunk. This interpretation is also compatible with our finding that reward-paired cues strongly increase production of press-approach chunks, given previous studies showing that habitual behavior is particularly sensitive to the motivational influence of such cues (Holland, 2004; Wiltgen et al., 2012).

As the reviewer notes, presses that were not followed by food-cup approach were particularly sensitive to reward devaluation. We have elaborated on our discussion of this issue (subsection “Inhibiting dopamine neurons spares the sensitivity of reward-seeking actions to reward devaluation”, last paragraph; Discussion, second paragraph, documents with colored text) and added a new analysis to help develop this finding (Figure 5D; Supplementary file 1G). However, we do not believe this is a defining feature of such actions. For instance, the variable-interval training protocol used in our PIT experiments is known to support habit formation. Thus, even discrete lever presses (without approach) performed during PIT tests were likely to have been insensitive to devaluation, had this been assessed. We suggest that such discrete lever presses are not exclusively aligned with either habitual or goal-directed control. Indeed, others have argued that repetitive lever pressing without subsequent food-cup approach may model OCD-like *compulsive checking* behavior, which we discuss in the last paragraph of the Discussion section.

Reviewer #2:[…] I have a series of issues with the manuscript:The terminology is very confusing, and the theoretical frame is globally difficult to grasp. There are many concepts that are used to capture very simple behaviors, even in the Results section (reward seeking, exploration, reward expectancy, motivation, vigor etc.). Could the authors make a set of simple predictions based on a simple theoretical frame and describe how they decide to operationalize the questions? There are places in the text when it is the case, but I'm afraid that there is a drift in the issues at stake between the Abstract and the Discussion.

We acknowledge that our original framework was confusing. As noted in our comments to reviewer #1, we have thoroughly revised the manuscript to correct this problem. Also, as noted above, we have adopted unambiguous terms for behavior in the Results section and figures, and have operationalized more complex or ambiguous terms.

The description of the methods is too superficial. It might be sufficient for specialists in the field (conditioning in rats) and/or people who know the work of this team, but not for a more general audience. Typically, including the contingencies and the timing of the reward schedules in the different tasks would help.

We apologize for any omissions and have added to this part of the Materials and methods section to make sure that these and other important details are specified.

I am having some trouble reconciling the conclusions of the authors and the data, as shown by the figure:Experiment 1: Again, the text is difficult to read but it seems to me that there are two things going on here: the operant behavior of these animals in the task is very 'habitual' (as opposed to goal-directed), since the rate of lever pressing decreases very little when reward is omitted (Figure 1D). In line with that, presumably, animals almost only try to collect the reward when they get a direct evidence (cue) that it has been delivered. So reward does not drive action and actions do not predict reward, and animals are left with Pavlovian processes to anticipate the reward. Is there more to it than that? If yes, it would require precise quantitative predictions and measures to demonstrate it. I would avoid the word 'exploration', unless the authors can demonstrate that animals try to get the reward by exploring another potential way to get the reward (e.g. another lever). Here, action seems to be driven much more by compulsion/ habits than exploration.

We generally agree with and have adopted this habit-based interpretation, as noted in our comments to reviewer #1. We have also avoided using exploration, exploratory reward seeking, and other ambiguous terms, as noted above.

Experiments 2 and 3:Looking at Figure 3B (Experiment 2) and Figure 4B and Figure 4—figure supplement 2 (Experiment 3), CNO alone has a strong effect on behavior and the difference between mCherry and hM4Di animals is really small. Not even sure it is significant, on the scale of the experiment. For example, in Experiment 2, Figure 3D shows a greater effect of CNO on response to CS- than CS+. Again, when looking at all the bars, I don't understand the conclusions.

Reviewers #2 and #3 both noted that systemic CNO may have had a nonspecific (hM4Di-independent) effect on lever pressing and that this should be acknowledged in the text. Consistent with this observation, our analysis of Total Lever Presses in Experiment 2 (Figure 3B) found a significant Drug x CS Period x CS Type interaction (*p* = 0.007), indicating that CNO partially suppressed lever pressing in a manner that was not necessarily limited to the hM4Di group. While this and related findings were indicated in the statistical output table for this analysis (Supplementary file 1A), we had not given it appropriate attention in the main text. We have edited the manuscript to acknowledge this potential nonspecific CNO effect (subsection “Inhibiting dopamine neurons during Pavlovian-to-instrumental transfer preferentially disrupts cue-motivated reward seeking, but not reward retrieval”, third paragraph). However, we also emphasize that this partial, nonspecific suppression of lever pressing does not account for the hM4Di-*dependent* effects of CNO that support our main conclusions. As noted in the Results section (see the aforementioned paragraph), we found a significant Group x Drug x CS period x CS type interaction (*p* =.002), indicating that the effects of CNO on PIT performance was more disruptive for the hM4Di group than for the mCherry group. Further analysis of data from the hM4Di group (excluding mCherry data) revealed a significant Drug x CS period x CS type interaction (p <.001). This group showed a CS+ specific increase in pressing during the vehicle test (CS period x CS type interaction, p <.001), but showed no such effect when tested on CNO (CS period x CS type interaction, p =.684). In contrast, analysis of data from the mCherry group (excluding hM4Di data) found a CS+ specific increase in lever pressing (CS Period x CS Type interaction, p <.001) that was not significantly altered by CNO (Drug x CS Period x CS Type interaction, p =.780). This null effect in the mCherry group is in line with a recently published study led by Dr. Kate Wassum (coauthor) that systemic CNO treatment does not significantly disrupt expression of the PIT effect in DREADD-free rats (Collins et al., 2019), which is cited in the current manuscript. It is also important to note that the same systemic CNO treatment did not significantly affect lever-press performance in Experiment 4 (Figure 5B). Based on these results and others noted below, we believe our original conclusions are strongly supported by the data, though we agree that they should be discussed in the context of potential nonspecific CNO effects.

We have also added an acknowledgement of the potential for CNO microinfusions to produce a partial nonspecific suppression in lever pressing in the mCherry rats used in Experiment 3B (subsection “Surgery”; Figure 4—figure supplement 2). Importantly, this effect did not reach significance (Drug * Site * CS Period * CS Type interaction, *p* =.068) and was driven by intra-mPFC (not NAc) CNO injections. Moreover, further analysis revealed that CNO did not significantly disrupt the CS+ induced increase in lever pressing (PIT score) in either group (*p*’s >.165). These findings indicate that the disruption of PIT detected in Experiment 3A (Figure 4C and 4D) following intra-NAc CNO injections in rats expressing hM4Di in VTA dopamine neurons resulted from local dopamine terminal inhibition and not a nonspecific action of that drug.

As detailed below, we believe that reviewer #2’s separate concern about effect sizes stems partly from variability in lever press rates across groups, drug conditions, and CS periods, which tended to obscure the effect of chemogenetic dopamine neuron inhibition on the increase in lever pressing that is specifically attributable to noncontingent CS+ presentation. In the next comment section, we develop the rationale behind our analysis of CS+ elicited behavior.

In Experiment 2, to test the prediction regarding the role of DA in PIT, they should test the interaction between 3 factors (CNO vs. Vehicle; mCherry vs. hM4Di and pre vs. post CS+). By the way, why using the pre/post CS+ when there is a CS- condition, which seems a priori as a better reference for assessing the specificity of the Pavlovian effects of the CS+, as opposed to general effect of having a stimulus? Looking at the data, CNO abolishes the increase in responding evoked by the CS+, and the major difference between mCherry and hM4Di is in the pre CS+ period… If the authors really want to take into trial to trial variability, they could use a relative measure (pre vs. post, on a trial by trial basis) but compare the effect of CS+ with the effect of CS- to assess PIT, and the influence of the various manipulations.

As requested by the reviewer, we have included a description of the significant Drug (CNO vs. Vehicle) x Group (mCherry vs. hM4Di) x CS period (pre vs. CS) interaction for CS+ trials (i.e., omitting CS- trials) (*t*(120) = -2.53, *p* =.013) in the Results section (subsection “Inhibiting dopamine neurons during Pavlovian-to-instrumental transfer preferentially disrupts cue-motivated reward seeking, but not reward retrieval”, third paragraph), which further supports the conclusion that inhibiting VTA dopamine neurons suppressed the ability of the CS+ to stimulate lever pressing.

We agree with the reviewer that the CS- is an essential control for evaluating the behavioral effects of the CS+ that specifically depend on that cue’s relationship with reward. However, it is not sufficient to simply contrast levels of responding during CS+ and CS- presentations without controlling for local baseline rates of responding during pre-CS periods. This is particularly important in PIT studies because lever press rates fluctuate sporadically due to the lack of reinforcement at test. For this reason, virtually all PIT studies control for pre-CS response levels (Holmes et al., 2010; Cartoni et al., 2016). It is for this reason that our primary analysis of PIT data always incorporates both CS Type (CS+ vs. CS-) and CS Period (pre-CS vs. CS) as factors, in addition to Drug (CNO vs. Veh) and Group (e.g., AAV group for Experiment 2 or injection site for Experiments 3A and 3B). When interpreting the results of these complex analyses, our main focus is on interactions involving *both CS Type and CS Period*, which reflects the ability for the CS+ but not the CS- to selectively increase responding relative to their cue-specific baseline levels. Main effects of Drug and Group are also of interest, as they represent general (cue-independent) behavioral effects. However, lower level interactions involving *only CS Period or CS Type* are harder to interpret because they are likely driven – to some degree – by incidental or nonspecific differences in pre-CS response rates, effects which are either uninteresting (random) or are better captured by the main effects of Drug or Group. Therefore, while the supplementary tables outline the full output of our main analyses (Supplementary file 1A-G), our discussion of results within the main text focuses on the results of most interest (i.e., main effects of Drug, Group, the Drug * Group interaction, or any interactions involving *both* CS Period and CS Type). We have modified the “Statistical Analysis” subsection of the Materials and methods to make this aspect of our data interpretation explicit to readers (third paragraph).

Incidental and/or nonspecific differences in pre-CS response rates can also make it difficult for readers to identify the source of the complex interactions expressed in raw data (total presses) in Figures 3B and 4B (each has 16 bars, including pre-CS periods). Therefore, we now include an analysis of *PIT scores* isolating CS+ induced changes in pressing (CS+ – pre-CS+; see Figures 3C, 4D, and Figure 4—figure supplement-2). This is a widely-used measure of the PIT effect (Pecina et al., 2006; El-Amamy and Holland, 2007; Bertran-Gonzalez et al., 2013; Laurent et al., 2016; Laurent et al., 2017; Alarcon et al., 2018; Panayi and Killcross, 2018) and facilitates data interpretation for general readers. This secondary analysis is also useful for confirming the source of potential drug effects on CS+ elicited behavior. We have used this analysis in place of the admittedly confusing and less compelling CNO suppression analysis that had been used in the original manuscript.

It is essentially the same thing for Experiment 3: the effect of CNO (here in the NAc) vs. vehicle is much bigger than the difference between mCherry and hM4Di. Ok, injecting CNO in the MFC has no effect, but this is potentially because the concentration of DA receptor (targeted by the CNO…) is smaller there than in the ACC.

This comment also raises another important point about our experimental design and statistical analysis, which depends heavily on within-subject comparisons (e.g., CS type, CS period, Drug). Not only is it crucial to consider baseline (pre-CS) response rates, as discussed above, it is essential to take into account unintended, presumably incidental, between-subject differences in PIT performance when evaluating the effect of CNO. For instance, in the Vehicle test of Experiment 2, rats in the hM4Di group showed a slightly stronger CS+ induced increase in lever pressing, relative to the mCherry group. These between-subject differences in off-drug PIT performance should be controlled for when assessing within-subject effects of CNO (Figure 3B). Similarly, for Experiment 3B (Figure 4—figure supplement 2), rats in the mCherry NAc group happened to show a relatively modest increase in lever pressing to the CS+ in their Vehicle test. Importantly, CNO administration did not significantly suppress this PIT effect and, in fact, was associated with a nonsignificant *increase* in CS+ elicited pressing. As noted above, this result indicates that the tendency for intra-NAc CNO administration to disrupt PIT performance in hM4Di expressing rats in Experiment 3A (see Figures 4C and 4D) was not due to a nonspecific drug effect.

One additional wrinkle to Experiment 3 is that Experiment 3B was conducted separately as a control study to analyze potential nonspecific CNO effects. While we used the same basic procedure as in Experiment 3A, direct comparisons between experiments are tempting but difficult to interpret. We believe that the critical question to ask when evaluating such data is whether CNO produced an effect in DREADD-free rats that accounts for the main findings, as is common in related studies (Augur et al., 2016; Laurent et al., 2016; Marchant et al., 2016; Campese et al., 2017; Lichtenberg et al., 2017; Alcaraz et al., 2018; Hsu et al., 2018; Collins et al., in press). Our findings demonstrate that this was not the case.

Separate issue, but as I mentioned above, describing what is essentially compulsive/habitual lever pressing as 'exploratory seeking' does not help.

As noted above, we have omitted this terminology.

Given these, I am not sure that any of the conclusions regarding the role of DA really stands. Again, the experimental design as a whole is excellent, I believe, and the results are interesting in terms of clarifying the neural and behavioral processes at stake in PIT, but ample revision are needed.

We thank the reviewer for these constructive comments and hope that our extensive revisions have adequately addressed their concerns.

Reviewer #3:This is a very interesting and cleverly-designed study that parses our how mesocorticolimbic dopamine (DA) transmission may be involved in the ability of Pavlovian cues to invigorate reward seeking, using a well-established Pavlovian-to-Instrumental transfer test. Using DREADD approaches to silence dopamine cell bodies they show that suppressing DA activity blocks or attenuates the PIT effect. Moreover, this appeared to be mediated primarily by DA activity in the accumbens, but not prefrontal cortex. What was particularly important about these findings is that the authors conducted a sophisticated microstructural analysis of behavior, focusing on how often rats may approach the food cup (as a measure of reward expectancy). This analysis showed that, even though reducing DA activity suppressed the increase in lever pressing induced by Pavlovian, reward-associated cues, it did not affect approaches to the food cup, suggesting that DA's role in this context is to enhance motivation, but not reward expectancy. They also showed that reducing DA activity did not influence how reinforcer devaluation altered responding.This is a well-designed study that has important implications for teasing out what DA does (and does not do) in modulating reward seeking. The analyses are sound, and the Discussion is fair and scholarly. I have no major issues with the paper, but a few relatively minor points the authors should attend to.1) Subsection “Pavlovian conditioning”, was a 10 Hz tone used? Or 10 kHz (I'm not sure rats can hear 10 Hz).

We thank the reviewer for spotting this mistake. We used a 2kHz pure tone that pulsated at 10Hz (0.1s on/0.1s off). We have corrected the Materials and methods section accordingly (subsection “Pavlovian conditioning”).

2) "After a session of Pavlovian conditioning, rats were given a 30-min extinction session" – based on the preceding section, it's unclear what happened. Did rats receive a Pavlovian session after the instrumental training and before the first test and then every subsequent one? This should be clarified.

We have revised this passage to clarify our training/retraining procedure (subsection “Pavlovian-to-instrumental transfer (PIT) test”). After the last session of instrumental training, rats were given a session of Pavlovian (CS+) training and a 30-min extinction session on the days prior to the first PIT test. At the end of this paragraph we clarify that: “before each new round of testing, rats were given two sessions of instrumental retraining (RI-60s), one session of CS+ retraining, and one 30-min extinction session, as described above.”

3) Subsection “Inhibiting dopamine neurons during Pavlovian-to-instrumental transfer preferentially disrupts cue-motivated reward seeking, but not reward retrieval”, last paragraph – the reader is referred to Figure 3E and F, but there is no F panel. Do they mean D and E?

We have corrected our panel labels. Thank you for drawing our attention to this.

4) In looking at the mCherry group in Figure 3, it appears that systemic CNO may have attenuated the PIT effect (although clearly not as much as it did in the hM4Di group). I'm not sure if the analysis look at this comparison, but it would be important to point out either way if CNO on its own has some effect on PIT.

As noted in our response to a similar comment by reviewer #2, we have revised the manuscript to acknowledge this potential nonspecific effect of CNO and provide evidence that this effect does not account for the attenuated CS+ elicited lever pressing produced by CNO in the hMD4i group (subsection “Inhibiting dopamine neurons during Pavlovian-to-instrumental transfer preferentially disrupts cue-motivated reward seeking, but not reward retrieval”, third paragraph).

5) The authors use an interesting correlational analysis to try explain the variation in the accumbens CNO data. However, looking at the histology, there is also considerable variation in the placements of infusions, some in core, some in shell. This warrants some mention, and perhaps another analysis seeing if the effects were stronger in animals with placements located in one NAc subregion vs. another.

We have looked closely at this issue. Unfortunately, variability in injector location did not help resolve variability in effect of CNO on PIT performance. This is not entirely surprising given previous findings that dopamine signaling in both core and shell may contribute to cue-motivated lever pressing (Lex and Hauber, 2008; Peciña and Berridge, 2013). We note this issue in the main text (subsection “Pathway-specific inhibition of dopamine projections to NAc, but not mPFC, disrupts cue-motivated reward seeking but not retrieval”, first paragraph).

Additional References:

Alarcon DE, Bonardi C, Delamater AR (2018). Associative mechanisms involved in specific Pavlovian-to-instrumental transfer in human learning tasks. Q J Exp

Psychol (Hove) 71:1607-1625.

Alcaraz F, Fresno V, Marchand AR, Kremer EJ, Coutureau E, Wolff M (2018). Thalamocortical and corticothalamic pathways differentially contribute to goal directed behaviors in the rat. *eLife* 7.

Augur IF, Wyckoff AR, Aston-Jones G, Kalivas PW, Peters J (2016) Chemogenetic

Activation of an Extinction Neural Circuit Reduces Cue-Induced Reinstatement of

Cocaine Seeking. J Neurosci 36:10174-10180.

Bertran-Gonzalez J, Laurent V, Chieng BC, Christie MJ, Balleine BW (2013). Learning related translocation of delta-opioid receptors on ventral striatal cholinergic interneurons mediates choice between goal-directed actions. J Neurosci 33:16060-16071.

Campese VD, Soroeta JM, Vazey EM, Aston-Jones G, LeDoux JE, Sears RM (2017). Noradrenergic Regulation of Central Amygdala in Aversive Pavlovian-to Instrumental Transfer. eNeuro 4.

Cartoni E, Balleine B, Baldassarre G (2016). Appetitive Pavlovian-instrumental Transfer: A review. Neurosci Biobehav R 71:829-848.

Colwill RM, Rescorla RA (1990). Effect of reinforcer devaluation on discriminative control of instrumental behavior. J Exp Psychol Anim Behav Process 16:40-47.

El-Amamy H, Holland PC (2007). Dissociable effects of disconnecting amygdala central nucleus from the ventral tegmental area or substantia nigra on learned orienting and incentive motivation. Eur J Neurosci 25:1557-1567.

Holmes NM, Marchand AR, Coutureau E (2010). Pavlovian to instrumental transfer: A neurobehavioural perspective. Neurosci Biobehav R 34:1277-1295.

Hsu TM, Noble EE, Liu CM, Cortella AM, Konanur VR, Suarez AN, Reiner DJ, Hahn JD, Hayes MR, Kanoski SE (2018). A hippocampus to prefrontal cortex neural

pathway inhibits food motivation through glucagon-like peptide-1 signaling. Mol

Psychiatry 23:1555-1565.

Laurent V, Chieng B, Balleine BW (2016). Extinction Generates Outcome-Specific

Conditioned Inhibition. Curr Biol 26:3169-3175.

Laurent V, Wong FL, Balleine BW (2017). The Lateral Habenula and Its Input to the Rostromedial Tegmental Nucleus Mediates Outcome-Specific Conditioned

Inhibition. Journal of Neuroscience 37:10932-10942.

Marchant NJ, Campbell EJ, Whitaker LR, Harvey BK, Kaganovsky K, Adhikary S, Hope BT, Heins RC, Prisinzano TE, Vardy E, Bonci A, Bossert JM, Shaham Y (2016). Role of Ventral Subiculum in Context-Induced Relapse to Alcohol Seeking after Punishment-Imposed Abstinence. J Neurosci 36:3281-3294.

Panayi MC, Killcross S (2018). Functional heterogeneity within the rodent lateral

orbitofrontal cortex dissociates outcome devaluation and reversal learning

deficits. *eLife* 7.

Pecina S, Schulkin J, Berridge KC (2006) Nucleus accumbens corticotropin-releasing factor increases cue-triggered motivation for sucrose reward: paradoxical positive incentive effects in stress? BMC biology 4:8.

Rescorla RA (1994) Transfer of Instrumental Control Mediated by a Devalued Outcome. Anim Learn Behav 22:27-33.

[Editors’ note: the author responses to the re-review follow.]

Reviewer #1:[…] This version of the paper includes additional control experiments, and the authors' new interpretations are much better supported by their data than previously. However, the authors should address a few additional points.1) The authors present evidence that the chunked operant-entry action sequence is resistant to devaluation. However, it is possible that port entry behavior is simply resistant to devaluation whether or not the entry follows an operant response. The authors show that animals make "noncontingent food cup approaches" (i.e., port entries), but do not analyze the rate of this behavior in any of their tasks. The argument that operant responses are resistant to devaluation because they are part of a chunked, habit action sequence would be strengthened if entries that are NOT part of such sequences were not resistant to devaluation.

We agree that the question of whether response-contingent and noncontingent approach responses differ in their sensitivity to reward devaluation is an important one. However, we are limited in our ability to analyze this issue in with the current data set. For our reward-specific devaluation, both rewards are retrieved from the same food cup. This makes it impossible to distinguish between noncontingent (press-independent) food cup approaches based on whether they are motivated by the devalued vs. non-devalued reward. We have indicated this reason for not including noncontingent approach analysis in the Results section (subsection “Inhibiting dopamine neurons spares the sensitivity of reward-seeking actions to reward devaluation”, last paragraph). We also note that other findings from our lab (not shown) from studies using a single action-outcome contingency task indicate that noncontingent approaches are indeed readily suppressed by reward devaluation, in contrast to press-contingent approaches. These findings are in line with previous reports that devaluation suppresses food-cup approach behavior (e.g., Balleine, 1992; Thrailkill and Bouton, 2017), including when those approaches are directly elicited by Pavlovian conditioned stimuli (Holland and Straub, 1979; Lichtenberg et al., 2017).

2) It would also be interesting to know whether noncontingent entries in the PIT test are dependent on dopamine/mesolimbic dopamine.

Our analysis of noncontingent entries during PIT testing (Figure 3—figure supplement 3, Figure 4—figure supplement 4) indicates that VTA dopamine inhibition (CNO in hM4Di expressing rats) does not disrupt this aspect of behavior.

3) Systemic CNO had a greater effect on PIT than intra-accumbens CNO. The authors should discuss the possible reasons for this.

We have added a passage (subsection “Pathway-specific inhibition of dopamine projections to NAc, but not mPFC, disrupts cue-motivated reward seeking but not retrieval”, second paragraph) discussing this issue.

4) Figure 4F and the similar supplementary figure are hard to understand. First, the X axis should be more descriptive (i.e., refer to CS-induced increase in lever presses without approach). Second, the Y axis reads "CNO suppression", but apparently greater suppression is represented by more negative numbers. This is counterintuitive. I think the authors are trying to say that the impact of CNO positively correlates with the impact of the CS on lever presses without approach, but they write that there is a negative correlation. The figure legend adds to the confusion by saying that the Y axis is "PIT Score for vehicle test – PIT Score for CNO test". If that's the case, then the lower PIT scores in CNO should result in positive values, not the predominantly negative values shown in the figure.

We agree that this figure panel and caption were generally confusing and have made several changes to clarify what is being presented. We continue to prefer using a measure (CNO-Vehicle) for which negative numbers reflect the degree of suppression is desirable. But we have corrected the caption and avoided directly using the term negative correlation in the main text (subsection “Pathway-specific inhibition of dopamine projections to NAc, but not mPFC, disrupts cue-motivated reward seeking but not retrieval”, fifth paragraph), which we agree may confuse readers. Importantly, the direction of the correlation statistic and nature of the measures used are clearly presented in the text and figures.